# Desaturase-dependent secretory functions of hepatocyte-like cells control systemic lipid metabolism during starvation in *Drosophila*

Jiayi Li[1], Kerui Huang[2], Indira Dibra[1], Ying Liu [2], Norbert Perrimon [2,3] & Matias Simons [1] ✉

Similar to the mammalian hepatocytes, *Drosophila* oenocytes accumulate fat during fasting, but it is unclear how they communicate with the fat body, the major lipid source. Using a modified protocol for prolonged starvation, we show that knockdown of the sole delta 9 desaturase, *Desat1* (SCD in mammals), specifically in oenocytes leads to more saturated lipids in the hemolymph and reduced triacylglycerol storage in the fat body as well as reduced survival. We further show that the insulin antagonist ImpL2 (IGFBP7 in mammals) is secreted from oenocytes during starvation in a Desat1-dependent manner. Flies with oenocyte-specific knockdown and overexpression of *ImpL2* exhibit higher and lower sensitivity to starvation, lower and higher triacylglycerol levels as well as higher and lower levels of *bmm* during starvation, respectively. Overall, this study highlights the importance of Desat1 in maintaining the proper functioning of oenocytes and the central role of oenocytes in the regulation of fat body lipid metabolism during periods of prolonged starvation.

Adaptation to changing environmental conditions is essential for survival. When animals adapt to nutrient shortage, intra-organ responses as well as inter-organ crosstalk mechanisms need to be mounted to coordinate whole-body energy homeostasis. A central organ in this process is the liver. In early stages of fasting, the glycogen stores of the liver are key for maintaining blood glucose levels. Upon prolonged fasting, there is a gradual transition in the body's energy source from carbohydrates to lipids reflected by the accumulation of lipids in the liver that are provided from lipolysis occurring in the adipose tissue. While the increase in circulating free fatty acids stimulates the production of ketone bodies in the liver, starvation also leads to major changes in the composition of circulating lipoproteins. In humans, for example, a shift towards more unsaturated triacylglycerol (TAG) and sterol esters in the serum has been observed[1]. All these processes are orchestrated by an array of hormones and other bioactive molecules such as hepatokines, adipokines or myokines[2].

*Drosophila melanogaster* has emerged as a major model system for the study of metabolic organ cross-talk[3]. In *Drosophila*, the fat body (FB) has been considered analogous to the human liver due to its roles in lipoprotein production and glycogen metabolism. However, its major function is to store fat similar to the adipose tissue. Oenocytes, on the other hand, were primarily associated with cuticular hydrocarbon and pheromone synthesis, until Gutierrez et al. demonstrated that oenocytes can accumulate lipids during periods of starvation similar to the liver[4,5]. Lipids accumulating in the oenocytes during fasting originate from the FB and are regulated by the insulin-like peptide, dILP6, which is secreted by the FB[6]. This behavior suggests a lipid exchange and crosstalk between the FB and oenocytes, analogous to the communication between adipose tissue and liver[7–9].

Why oenocytes store lipids during fasting is not understood. While one reasonable assumption is that, also in flies, lipids are used for ketogenesis, lipids may also be modified and stored as TAG by the

[1]Nephrogenetics Unit, Institute of Human Genetics, University Hospital Heidelberg, Heidelberg, Germany. [2]Department of Genetics, Blavatnik Institute, Harvard Medical School, Boston, MA, USA. [3]Howard Hughes Medical Institute, Harvard Medical School, Boston, MA, USA. ✉ e-mail: matias.simons@med.uni-heidelberg.de

oenocytes before being released again into the circulation. Similar to humans, the modifications could involve lipid unsaturation. In this regard, it is interesting that *Desat1* is highly expressed in oenocytes[4,10]. Desat1, the sole delta9 desaturase in *Drosophila* (corresponding to stearoyl-CoA desaturases (SCDs) in mammals)[11], converts saturated into monounsaturated fatty acids and has previously been suggested to promote hydrocarbon formation in oenocytes[4]. However, as efficient TAG formation depends on SCDs in mammalian cells[12,13], lipid modification by oenocyte Desat1 may also promote systemic TAG storage, which has been shown to be essential for survival under fasting conditions[14,15].

Like in mammals, important vehicles for lipids in the hemolymph, the fly's circulatory system, are lipoproteins. The apoB-family lipoprotein, Lipophorin (Lpp), is synthesized and secreted by the FB and serves as the primary lipid carrier in the hemolymph. Under normal feeding conditions, Lpp transport to the midgut is crucial where it is loaded with sterols, medium-sized diacylglycerol (DAG) and phosphatidylethanolamine (PE) obtained by de novo–synthesis or from the diet. Lpp loading on the enterocyte surface depends on another FB-derived apoB-family lipoprotein, Lipid Transfer Particle (LTP), which enters the enterocytes via endocytosis while Lpp remains bound to the surface[16]. In the wing disc, on the other hand, the low-density lipoprotein (LDL) receptor orthologues LpR1/2 promote Lpp internalization, while also mediating non-endocytic lipid uptake mechanisms[17,18]. These examples illustrate the different modes of interactions of Lpp with its target cells. How lipoproteins mediate the transfer of lipids from the FB to oenocytes and vice versa in fasting conditions is unclear.

Here, we use *Drosophila* to highlight the role of oenocytes and Desat1 in the starvation response. We show that prolonged starvation causes alterations in the circulating lipidome and affects lipid storage and release of lipoproteins by oenocytes. Desat1 in oenocytes is essential for inducing these lipidomic changes by promoting lipid flux through the TAG compartment in oenocytes and by controlling lipid storage in the FB. The latter is facilitated by the secretion of ImpL2 that controls FB lipolysis and promotes starvation resistance.

## Results

### Lipidomic shifts in *Drosophila* during nutrient restriction

In this study, we established a modified starvation medium for *Drosophila* which is based on the omission of all calories in holidic medium (besides 10 kcal/L stemming from the acetic acid to match the pH of natural food sources)[19]. Compared to the commonly used phosphate-buffered saline (PBS)-based starvation media, the medium contains a number of specific ions and metals in a 2% agar medium (see methods and Supplementary Table 5). As demonstrated in Fig. 1a, b, a comparative analysis of different starvation protocols revealed that flies with our starvation medium exhibited a significantly extended survival, allowing us to detect metabolic changes over a time period of up to 10 days. Survival extension was more pronounced in females (Fig. 2a, b) and was largely independent of the calories derived from acetic acid supplementation (Supplementary Fig. 1a). We performed lipidomics on whole fly and hemolymph samples at ad libitum fed conditions as well as on different days of nutrient restriction (NR). In the whole fly samples (female), TAG levels exhibited a modest elevation at the beginning of NR, followed by a decline in the subsequent stages (Fig. 1d). Conversely, levels of PE, phosphatidylcholine (PC) and phosphatidylinositol (PI) increased markedly throughout the starvation period in whole fly samples (Fig. 1d). In hemolymph samples, we observed decreases in DAG, PE and TAG during starvation (Fig. 1e). We further observed an overall increase in the number of double bonds and length of acyl chains within several lipid classes in both whole-fly and hemolymph samples (Fig. 1f, i and Supplementary Fig. 1b-i). In the hemolymph, the biggest double bond increase was found for DAG and PE. Hence, an overall increase in the number of double bonds and lipid

chain length in circulating lipoproteins represents the main change in lipid composition during NR.

### Oenocytes control circulating lipid and global TAG levels during starvation in a Desat1-dependent manner

A remarkable hepatocyte-like feature of oenocytes is starvation-induced steatosis, which can be seen both in larval and adult stages (Fig. 1j, k). As *Desat1* is highly expressed in these specialized cells[4,10], we hypothesized that the lipid alterations we observed during starvation may be related to Desat1-dependent lipid storage and release by oenocytes. To study this, we first performed lifespan experiments using flies with or without inducible oenocyte-specific *Desat1* knock-down (KD). As shown in Fig. 2a, b, we indeed found a higher starvation sensitivity in both female and male flies with GAL80-dependent temperature-sensitive expression of *Desat1* RNAi[10] in oenocytes (hereafter referred to as *OE^ts>Desat1^RNAi*) as well as RU486-dependent expression of *Desat1* RNAi compared to flies expressing an empty vector RNAi construct in oenocytes (*Control^RNAi*) or without RU486 at 25 °C, respectively (Supplementary Fig. 2a). As additional controls in the GAL80 experiments, we used *GFP* RNAi (Fig. 7a) and maintained the expression of the GAL80 repressor at 18 °C (Fig. 2c). Both of these controls did not show any significant differences in lifespan during NR.

Cold exposure is another stress condition that involves a global shift towards decreased lipid saturation levels to counteract the decreased membrane fluidity, also referred to as homeoviscous adaptation[20,21]. To test whether Desat1 in oenocytes contributes to this global response, we exposed flies to 4 °C for 24 h (male) or 48 h (female) and then measured the number of flies that are able to recover. *OE^ts> Control^RNAi* flies showed a greater resistance to 4 °C exposures compared with *OE^ts> Desat1^RNAi* flies (Fig. 2d), providing further support that oenocyte Desat1 is important for systemic lipid metabolism in adaptation to starvation and cold stress.

To directly study the impact of oenocyte Desat1 on lipid compositions, we measured the global and hemolymph lipidomes of *OE^ts>Desat1^RNAi* and control flies under normal and NR conditions. As shown in Fig. 2e, f, the saturation level of several major lipids in the hemolymph from *OE^ts>Desat1^RNAi*, particularly in DAG and PE, were elevated compared to the control group in both fed and starved conditions. By contrast, we observed only modest changes in the number of double bonds in TAGs and all glycerophospholipids (GPLs) in whole fly samples (Fig. 2g, h). However, the differences of all lipids in both whole fly and hemolymph between the OE^ts>*Control^RNAi* and *OE^ts> Desat1^RNAi* groups appeared more pronounced in NR compared to fed conditions (Supplementary Fig. 2b, c). In particular, the starvation-induced increase of acyl chain length of most hemolymph lipids was suppressed in the *OE^ts>Desat1^RNAi* group (Fig. 2i, j), altogether indicating that Desat1 function in oenocytes controls both saturation level and length of the circulating lipidome. We further noticed that lipoprotein levels, as reflected by hemolymph DAG and PE levels, dropped during starvation, and that this drop was much faster in the *OE^ts>Desat1^RNAi* group (Fig. 3a). This was accompanied by a decrease in the global TAG levels in the *OE^ts> Desat1^RNAi* group. Particularly, on the second day of starvation, there was a dramatic decrease in TAG levels compared with the *OE^ts>Control^RNAi* group (Fig. 3b).

In summary, these findings suggest that oenocytes are capable of processing and releasing lipids into the hemolymph as well as controlling both global TAG levels in a Desat1-dependent manner during NR. Considering the importance of these energy stores during starvation, this rapid lipid utilization likely explains the higher starvation sensitivity of *OE^ts>Desat1^RNAi* flies.

### Oenocytes regulate FB lipid storage in a non-autonomous manner

To assess the TAG dynamics during the NR period, we performed BODIPY staining at day 3, 5, and 7 of NR. In *OE^ts>Control^RNAi* flies, lipid

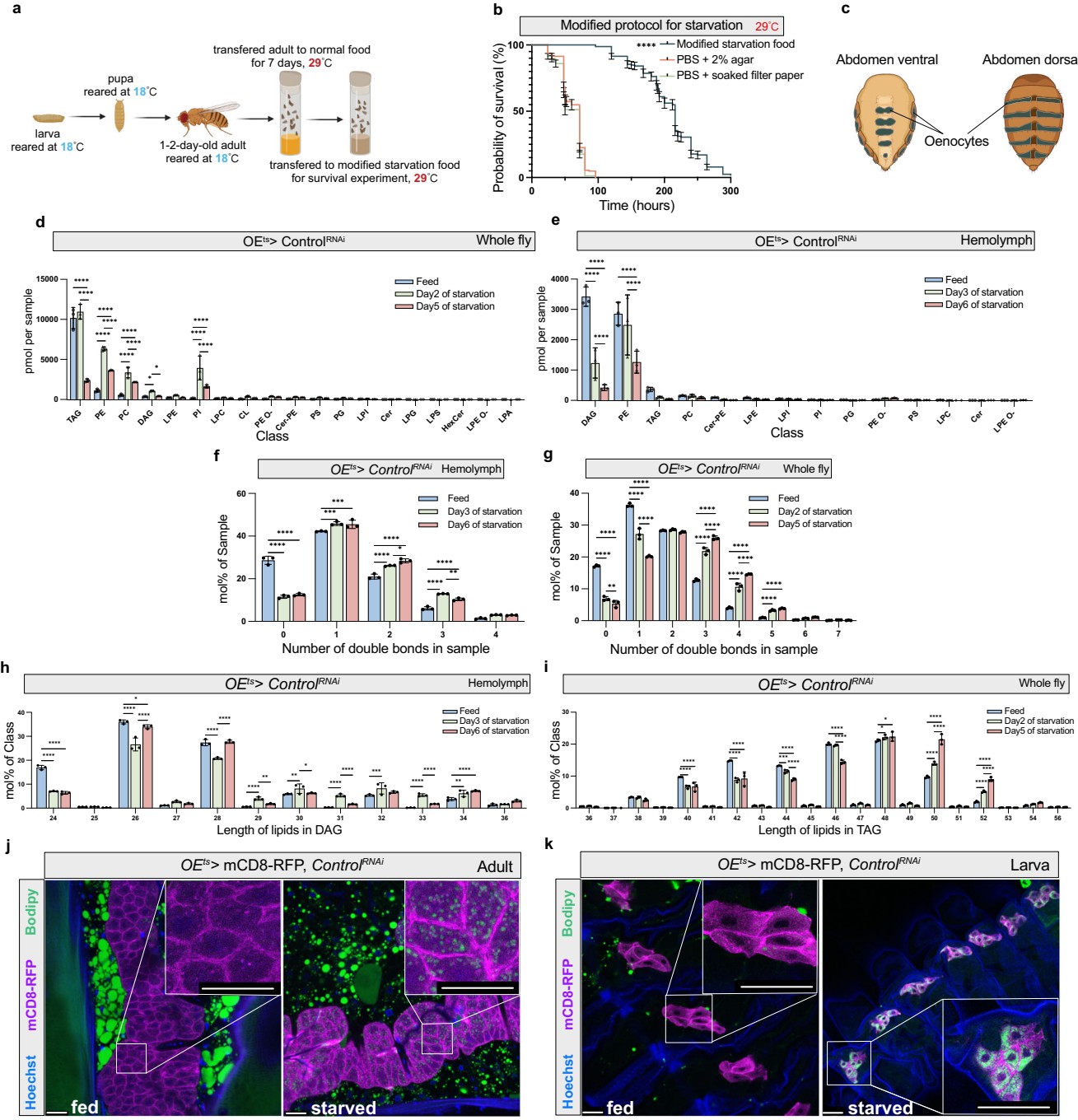

**Fig. 1 | Lipidome shifts in adult *Drosophila* during starvation. a** A protocol for starvation assay used for adult fly with Gal80[ts] driver at 29°C. **b** Starvation sensitivity assay using a modified protocol for long-term starvation experiments, $n \approx 120$ adult female flies (PromE-Gal4; *tub*-Gal80[ts]; *Control*[RNAi] (here termed *OE*[ts>]*Control*[RNAi])). Data are presented as mean values ± SEM. $P < 1 \times 10^{-15}$, *p* value was calculated using Log-rank (Mantel-Cox) test. PBS, Phosphate Buffered Saline. **c** Schematic of the localization of oenocytes in adult flies. **d**, **e** Quantitative lipidomic analysis of hemolymph or whole fly samples. Error bars indicate standard deviation, $n = 3$. Statistical tests: two-way ANOVA with Tukey's multiple comparisons test. Data are presented as mean values ± SD. **f**, **g** The number of double bonds in hemolymph or whole fly samples at different days of starvation, $n = 3$. Statistical

tests: two-way ANOVA with Tukey's multiple comparisons test. Data are presented as mean values ± SD. **h**, **i** Length analysis of major lipids class from hemolymph or whole fly samples, $n = 3$. Statistical tests: two-way ANOVA with Tukey's multiple comparisons test. Data are presented as mean values ± SD. **j**, **k** Lipid droplets (LD) were visualized with BODIPY (green) during modified starvation protocol in adult flies and during PBS starvation in larvae. Cell outlines were marked with mCD8-RFP (red). Scale bars, 20 µm. *, $P < 0.05$; **, $P < 0.01$; ***, $P < 0.001$; ****, $P < 0.0001$. Error bars indicate standard deviation. Source data for plots and exact p-value are provided as a source data file, **a** and **c** created in BioRender.com. Li, J. (2025) (https://BioRender.com/096lme4; https://BioRender.com/3p1saxj) is licensed under CC BY 4.0.

droplet (LD) accumulation in oenocytes (marked with co-expression of CD8-RFP) was increased during the early and middle phases of starvation, with a noticeable decrease in the later stages. Interestingly, the adjacent FB began to exhibit small LDs beyond day 5, suggesting a

bidirectional lipid exchange between oenocytes and FB (Fig. 3c, d). In *OE*[ts>] *Desat1*[RNAi] flies, we detected significantly reduced LD formation at all stages of starvation (Fig. 3e), in line with previous studies on the role of SCDs in TAG synthesis[12,13]. Similarly, when silencing *Desat1* only in

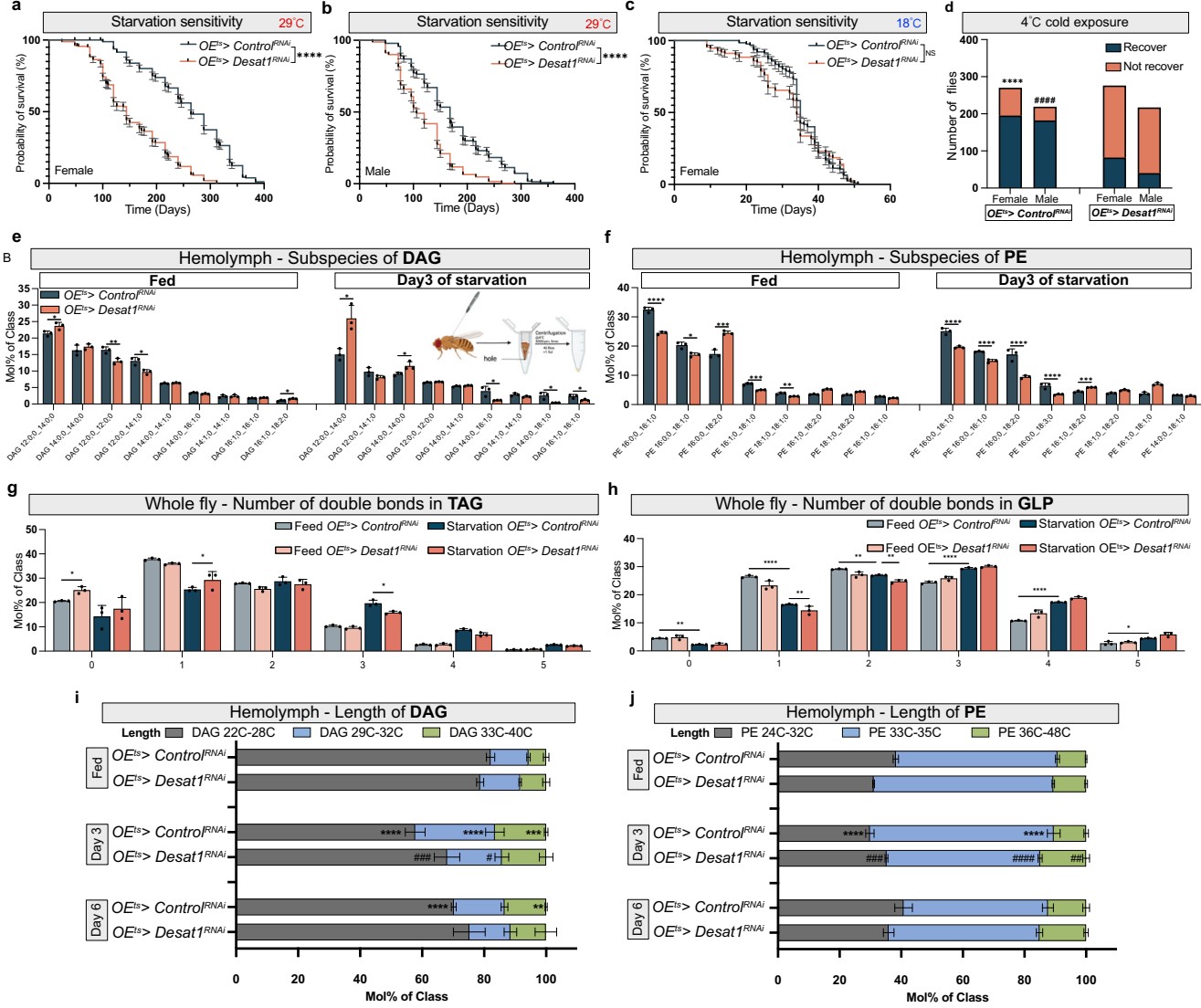

**Fig. 2 | OE^ts^>Desat1^RNAi^ flies exhibit altered lipid profiles and higher sensitivity to starvation and cold stress. a, b** $OE^{ts}$>$Desat1^{RNAi}$ adult flies show higher starvation sensitivity at 29°C compared with $OE^{ts}$> $Control^{RNAi}$ flies in both male and female, $n = 270$ in female flies and $n = 238$ in male flies. $P$ value was calculated using Log-rank (Mantel-Cox) test. **c** $OE^{ts}$>$Desat1^{RNAi}$ adult female flies show no different in lifetime during starvation compared with control group at 18°C, $n = 113$. $P$ value was calculated using Log-rank (Mantel-Cox) test. **d** Both $OE^{ts}$>$Desat1^{RNAi}$ adult female and male flies show a better recovery from cold exposure. 24 h cold exposure for male adult flies and 48 h for female flies, $n \approx 217$ for male, n ≈ 270 for female. P value was calculated using two-sided Chi-square, ****($P < 1 \times 10^{-15}$): female $OE^{ts}$>$Desat1^{RNAi}$ vs. $OE^{ts}$>$Control^{RNAi}$ flies, ####($P < 1 \times 10^{-15}$): male $OE^{ts}$> $Desat1^{RNAi}$ vs. $OE^{ts}$>$Control^{RNAi}$ flies. **e, f** Subspecies lipid analysis (diacylglycerol (DAG) and phosphatidylethanolamine (PE)) of hemolymph samples between $Desat1$ KD and control groups in fed or starvation conditions, $n = 3$. Statistical tests: multiple two-tailed unpaired $t$ tests.

*, $P < 0.05$; **, $P < 0.01$; ***, $P < 0.001$; ****, $P < 0.0001$. **g, h** The number of double bonds in triacylglycerol (TAG) and GLP (glycerophospholipids) from flies' samples in fed or starvation conditions, $n = 3$. Statistical tests: two-way ANOVA with Tukey's multiple comparisons test. *, $P < 0.05$; **, $P < 0.01$; ****, $P < 0.0001$. **i, j** Changes in the percentage of different chain lengths of DAG and PE in hemolymph during starvation, $n = 3$. Statistical tests: two-way ANOVA with Tukey's multiple comparisons test. #: $OE^{ts}$>$Desat1^{RNAi}$ flies at day 3 of starvation vs. $OE^{ts}$>$Control^{RNAi}$ flies at day 3 of starvation. *: $OE^{ts}$>$Control^{RNAi}$ flies at day 3 or day 6 of starvation vs. $OE^{ts}$>$Control^{RNAi}$ flies at feed condition. *, $P < 0.05$; **, $P < 0.01$; ***, $P < 0.001$; ***, $P < 0.0001$. ##, $P < 0.01$; ###, $P < 0.001$; ####, $P < 0.0001$. Error bars indicate standard deviation. Data in (**a–c**) are presented as mean values ± SEM. Data in (**e–j**) are presented as mean values ± SD. Source data for plots and exact $p$-value are provided as a source data file. **e** created in BioRender. Li, J. (2025) (https://BioRender.com/5t3oi9n) is licensed under CC BY 4.0.

the FB, a strong suppression of LD formation in the FB as well as global TAG content was observed. This phenotype was coupled with a strong developmental arrest at larval stages (Supplementary Fig. 3a-f). Silencing $Desat1$ in the FB only during adult stages also led to LD and TAG reduction as well as reduced survival during starvation compared to control (Supplementary Fig. 3g-h,d).

An important additional finding in $OE^{ts}$> $Desat1^{RNAi}$ flies was the non-autonomous effect on the FB, specifically during NR. The surrounding FB exhibited a sharp decline of BODIPY staining at the onset of starvation, suggesting an accelerated lipolysis in the FB compared to the control (Fig. 3e) during starvation. For systemic

effects, we also studied adipose tissue from the head of the flies and found a drop in TAG levels in $OE^{ts}$> $Desat1^{RNAi}$ flies (Supplementary Fig. 3i). Interestingly, this correlated with an upregulation of *brummer* (*bmm*), the main neutral TAG lipase (ATGL in mammals), suggesting that oenocytes might regulate global TAG levels by controlling *bmm* in the FB during starvation (Fig. 3f, g). To rule out that global TAG depletion is after all the result of enhanced lipolysis in the oenocytes, we first co-silenced $Desat1$ and $bmm$ in oenocytes and monitored TAG levels over the course of starvation. Intriguingly, akin to the $OE^{ts}$> $Desat1^{RNAi}$ group, the whole-body TAG levels in the $Desat1/bmm$ double KD group still exhibited a significant drop by day

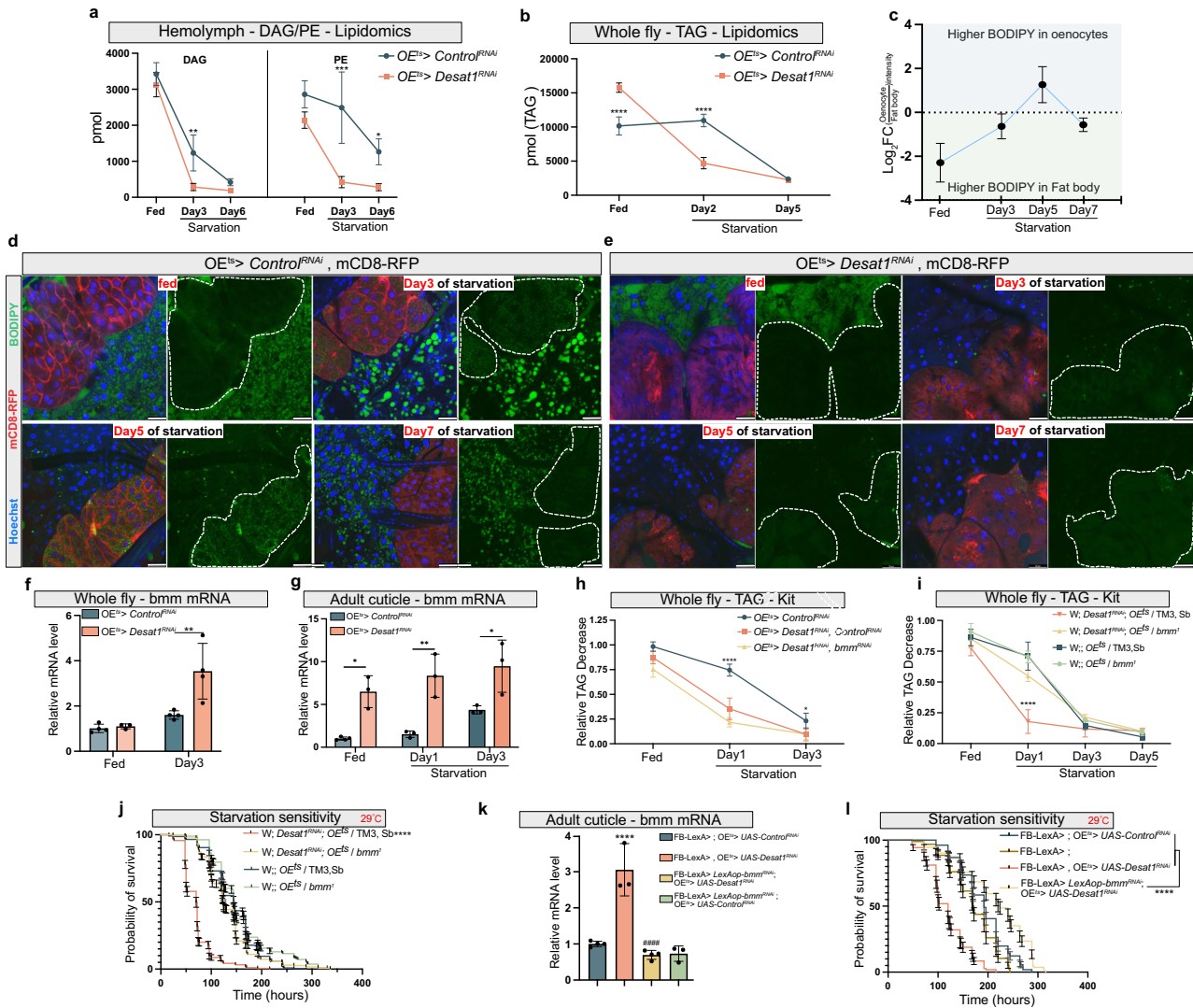

**Fig. 3 | OE$^{ts}$ > Desat1$^{RNAi}$ adult flies exhibit a faster lipid consumption in the fat body during starvation. a** The amount of DAG and PE at different days of starvation in hemolymph between OE$^{ts}$> Desat1$^{RNAi}$ adult female flies and OE$^{ts}$>Control$^{RNAi}$ adult female flies measured by mass spectrometry, $n = 3$. Statistical tests: two-way ANOVA. **b** The amount of TAG at different days of starvation in whole fly sample between OE$^{ts}$>Desat1$^{RNAi}$ and OE$^{ts}$>Control$^{RNAi}$ adult female flies measured by lipidomics, $n = 3$. Statistical tests: two-way ANOVA. **c** The BODIPY intensity in oenocyte or surround fat body during different days of starvation. **d, e** LD staining (green) at different days of starvation. mCD8-RFP (red) was overexpressed in oenocytes to mark the cell membrane and dashed lines showed the oenocytes. Scale bars: 20 µm. **f, g** bmm mRNA level in whole fly sample or cuticle at different day of starvation, $n = 3$. Statistical tests: two-way ANOVA. **h** TAG level at different days of starvation in control, Desat1 KD and double KD of Desat1 and bmm specifically in oenocytes measured by TAG kit, $n = 4$. Statistical tests: two-way ANOVA with Tukey's multiple comparisons test. *: OE$^{ts}$>Control$^{RNAi}$ vs. OE$^{ts}$>Desat1$^{RNAi}$ or

OE$^{ts}$> Desat1$^{RNAi}$, bmm$^{RNAi}$ adult female flies. **i** TAG level at different days of starvation measured by TAG kit, $n = 3$. Statistical tests: two-way ANOVA with Tukey's multiple comparisons test. ****: $P < 0.00001$, Desat1 KD group vs. all other groups.
**j** Starvation sensitivity assay between control flies, Desat1 KD flies and Desat1 KD flies carrying one allele of the bmm loss-of-function mutation, bmm$^1$, $n = 180$. $P$ value was calculated using Log-rank (Mantel-Cox) test. ****, $P < 0.00001$(w; Desat1-RNAi; PromE, gal80ts/Sb,TM3 vs. all other groups). **k** bmm mRNA level from adult cuticle samples with or without oenocytes Desat1 KD to validate the knockdown efficiency of LexAop-bmm RNAi, $n = 3$. ####: flies with fat body bmm KD and oenocytes Desat1 KD vs. flies with oenocytes Desat1 KD) ****: Control group vs. flies with oenocytes Desat1 KD). Statistical tests: one-way ANOVA. **l** Starvation assay between control flies (FB-LexA; FB-LexA >, OE$^{ts}$>Control$^{RNAi}$) and fat body-specific bmm KD (FB-LexA>LexAop-bmm RNAi) with or without oenocytes Desat1 KD flies (OE$^{ts}$>Desat1 RNAi). $n = 110$, ****, $P < 1 \times 10^{-15}$. Error bars indicate standard deviation. Source data for plots and exact $p$-value are provided as a source data file.

2 of starvation (Fig. 3h), despite the accumulation of LDs within the oenocytes (Supplementary Fig. 4a, b).

To test for the importance of global lipolysis reduction for survival in starvation, we next removed one copy of bmm in OE$^{ts}$> Desat1$^{RNAi}$ flies. Remarkably, this genetic strategy rescued the increased starvation sensitivity of OE$^{ts}$> Desat1$^{RNAi}$ flies and also correlated with a suppression of global TAG consumption in OE$^{ts}$> Desat1$^{RNAi}$ animals (Fig. 3i, j). To study FB-specific lipolysis, we used the LexA/LexAop system as an alternative binary expression system to downregulate bmm specifically in fat bodies (using r5-LexA driver) of OE$^{ts}$> Desat1$^{RNAi}$ flies. With this approach, bmm induction in OE$^{ts}$> Desat1$^{RNAi}$ flies was

significantly suppressed, which led to a strong lifespan increase during starvation (Fig. 3k, l). Together with the strong LD decline in the FB of OE$^{ts}$> Desat1$^{RNAi}$ animals, this indicates that oenocytes non-autonomously regulated the dynamic of lipids in the FB by suppressing lipolytic activity during starvation.

**Desat1 deficiency causes an accumulation of lipoproteins and actin on the oenocyte surface**

To further explore the mechanisms by which oenocytes influence hemolymph lipid composition and systemic lipid metabolism, we focused on lipoproteins. For lipoprotein visualization, we used a

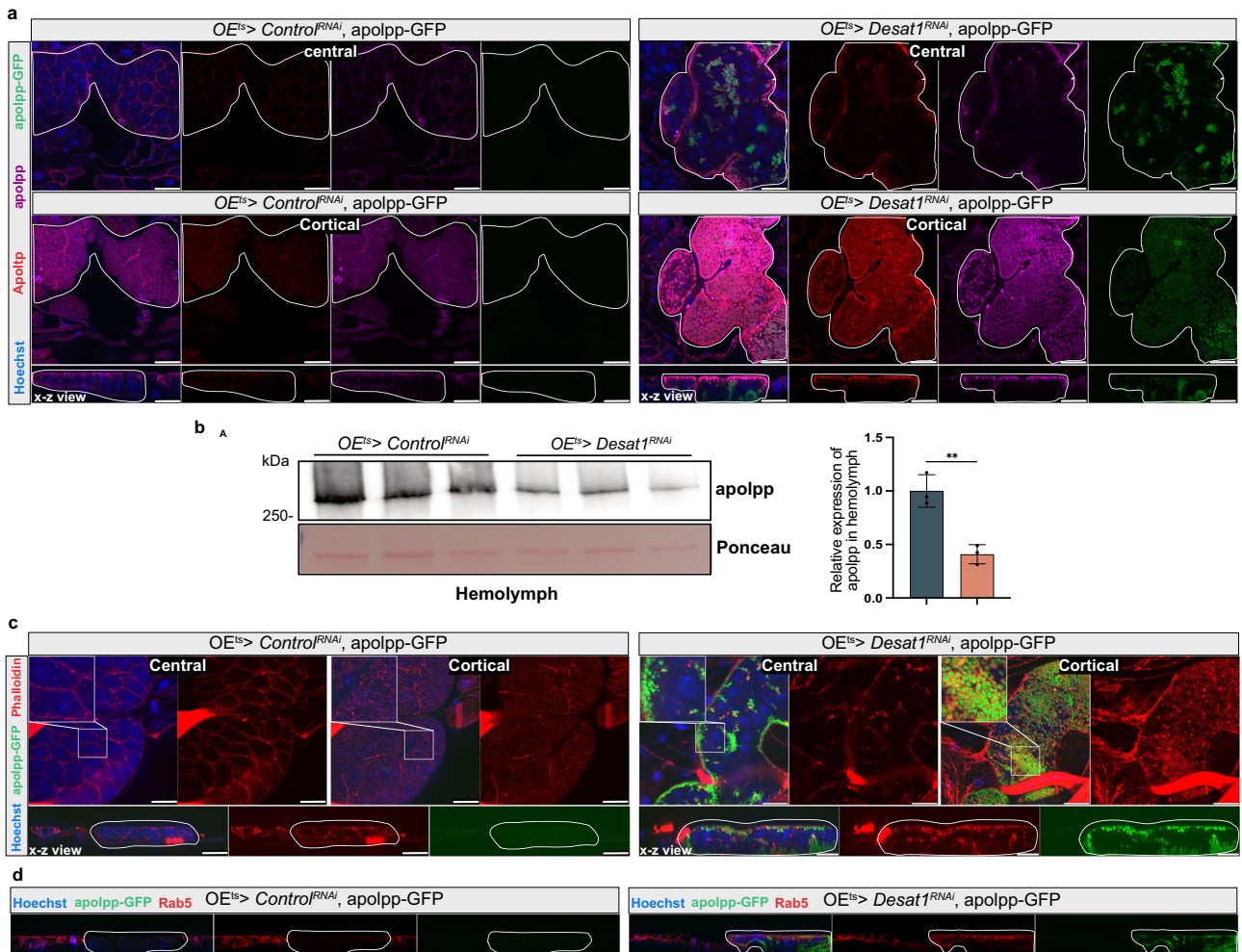

**Fig. 4 | $OE^{ts}$ > $Desat1^{RNAi}$ flies show lipoprotein sequestering in the oenocyte cortex. a** Representative immunofluorescence images of lipoproteins in central or cortical sections of oenocytes with or without *Desat1* KD. Oenocytes with *Desat1* KD exhibit a strong lipoprotein accumulation in the cortex region. X-z view displayed a vertical cross-section along the X-axis and dashed lines showed the oenocytes. Note that the gain in the *Desat1* KD group (right panel) was decreased by around 40% to avoid the saturation of image. Also note that while ApoLpp-GFP (detected without antibody) also accumulates it does not fully overlap with anti-apolpp and anti-Apoltp, suggesting cleavage of the GFP tag or issues with antibody penetration in the *Desat1* KD oenocytes. *n* > 4, scale bars: 20μm. **b** Western blot analysis of apolpp

in hemolymph from control and *Desat1* KD group and quantification of hemolymph apolpp level in hemolymph. Only one band was seen after Ponceau staining, and this was used for normalization. *n* = 3. Statistical tests: two-tailed unpaired *t* test. **c** Representative immunofluorescence images of phalloidin (red) and endogenous apolpp-GFP (green) in central or cortical sections of of oenocytes with or without *Desat1* KD. *n* = 3, scale bars: 20 μm. Data are presented as mean values ± SD. **d** X-z view of Rab5 staining and apolpp-GFP with and without *Desat1* KD. Dashed lines mark the oenocytes, *n* = 3, scale bars: 20 μm. Source data for plots and exact *p*-value are provided as a source data file.

transgenic line in which apolpp is endogenously tagged with a super-folder green fluorescent protein (sfGFP) at the C-terminus, along with apolpp and Apoltp antibodies for immunofluorescence studies. Intri-guingly, a pronounced accumulation of both apolpp and Apoltp was detected on the surface of oenocytes in the $OE^{ts}$> $Desat1^{RNAi}$ group, as shown in Fig. 4a. The same was true for the sfGFP signal, although sfGFP and antibody staining in the Desat1 KD group did not fully overlap (Fig. 4a), suggesting potential issues with antibody penetration or partial cleavage of the sfGFP tag. We also performed Western blot analysis using hemolymph samples to assess the levels of lipoproteins in the $OE^{ts}$> $Desat1^{RNAi}$ group (Fig. 4b). Consistent with the lipidomics results, there was a marked decrease in apolpp levels in the hemo-lymph of the $OE^{ts}$> $Desat1^{RNAi}$ group (Fig. 4b). Moreover, in fed condi-tions, an accumulation of lipids in the gut could be observed, suggesting that lipoproteins were indeed sequestered by the oeno-cytes upon *Desat1* silencing (Supplementary Fig. 5a).

To rule out the possibility that the lipoproteins were after all synthesized in oenocytes, we silenced *apolpp* in oenocytes. As shown

in Supplementary Fig. 5b, the apolpp accumulation on the surface of oenocytes in the double KD of *Desat1* and *apolpp* was comparable to the one seen with the single *Desat1* KD. In addition, silencing of *apolpp* in oenocytes did not cause any sensitivity to starvation (Supplemen-tary Fig. 5c), altogether suggesting that lipoproteins trapped on the surface of *Desat1*-deficient oenocytes do not originate from oenocytes and most likely are FB-derived.

Imaging with higher magnification revealed that apolpp-GFP accumulated in vesicles close to the oenocyte surface (Fig. 4c). Accordingly, the surface marker CD8-RFP showed a patchier dis-tribution as well as a more restricted diffusion on the surface of *Desat1* KD oenocytes compared to the control, as determined by FRAP (fluorescence recovery after photobleaching) experiments (Supple-mentary Fig. 6). Co-staining with phalloidin and an antibody against the early endosomal marker Rab5 showed that both actin filaments and Rab5 were markedly increased compared to the control and appeared to encapsulate the cortical apolpp vesicles (Fig. 4c, d). Moreover, *Rab5* KD led to a similar accumulation of apolpp and actin (Supplementary

Fig. 7), suggesting that impaired endocytosis of lipoproteins might be linked with the actin accumulation. Accordingly, the KD of the recycling endosomal marker *Rab11* caused a similar, albeit milder, Lpp phenotype as *Rab5* depletion, while this was not the case when silencing the late endosomal *Rab7* (Supplementary Fig. 8) or the autophagosomal *Atg1* (Supplementary Fig. 9).

Based on these findings, we reasoned that a decrease in actin accumulation might be able to release lipoproteins from the oenocytes into the hemolymph. Therefore, we attempted to depolymerize the excessive F-actin by overexpressing *tsr (twinstar)* or knocking down *Limk1* (LIM domain kinase 1)[22–24]. Remarkably, both treatments not only led to a suppression of the actin accumulation (Fig. 5a) but also to a strong decrease of the apolpp accumulation on the oenocyte surface (Fig. 5b) as well as a normalization of lipoprotein levels in the hemolymph, TAG level and starvation sensitivity (Fig. 5c, f).

Altogether, these results demonstrate that impaired lipoprotein internalization is associated with a rigid meshwork of actin and endosomal vesicles in the oenocyte cortex and that lowering F-actin in oenocytes not only normalizes lipoproteins in the circulation but also lipolysis and starvation sensitivity in *OE^{ts}>Desat1^{RNAi}* animals.

## Single-nucleus RNA-seq identifies candidate proteins secreted from oenocytes during nutrient restriction

To test how the alterations of the oenocyte cortex in *Desat1* KD cells might impair exocytosis, we overexpressed albumin-mCherry, whose secretion into the hemolymph can be monitored in pericardial nephrocytes expressing cubilin for albumin uptake[25]. A notable accumulation of albumin-mCherry within the oenocytes was accompanied by a diminished mCherry signal in the pericardial nephrocytes in the *OE^{ts}>Desat1^{RNAi}* group compared to the control group (Fig. 6a, b). This result suggested that oenocytes with *Desat1* KD exhibit impaired protein secretion capabilities. As the early secretory pathway appeared morphologically normal (Supplementary Fig. 10), the secretion defects may be caused by the cortex alterations, but other reasons are also possible. To further validate this effect in another cell type, we silenced *Desat1* in the FB and examined the secretion of apolpp. Also in this tissue, we observed a strong accumulation of apolpp within the FB (Supplementary Fig. 11), providing additional evidence that proper Desat1 function is essential for normal cellular secretion.

To identify secreted factors relevant in the tissue crosstalk during starvation, we utilized single-nucleus RNA sequencing (snRNA-seq) to identify genes encoding secreted proteins that are altered during starvation. 90 adult flies (excluding the head) at various stages of starvation (fed state, day 2, and day 5 of starvation) were used for snRNA-seq. 25 Cell clusters were visualized using a uniform manifold approximation and projection (UMAP) plot (Fig. 6c). The genes that changed significantly in oenocytes during starvation compared to the fed condition are presented in a volcano plot in Supplementary Fig. 12a. To further elucidate the metabolic pathways activated in oenocytes during starvation, we performed a gene set enrichment analysis using the online tool PANGEA[26]. The results indicated that genes altered at day 2 of starvation were predominantly enriched in gene ontology (GO) sets related to fatty acid metabolism (Fig. 6d). Our findings therefore support the hypothesis that oenocytes actively engaged in lipid processing during periods of starvation.

## ImpL2 secretion by oenocytes controls starvation responses

Using the algorithm shown in Supplementary Fig. 12b, we then selected 33 genes that were annotated as secreted proteins and either changed during starvation at the mRNA level in oenocytes or were mainly expressed in the oenocytes based on Fly Cell Atlas[27]. From these 33 genes, we selected 8 genes and measured whole-body TAG levels during starvation upon KD in oenocytes (Supplementary Fig. 12c). As flies with oenocyte-specific silencing of *ImpL2* showed the most significant TAG downregulation at day 3 of starvation, we focused on this

gene. The secreted protein ImpL2 was mainly expressed in oenocytes and its expression in oenocytes increased during starvation (Fig. 6e and Supplementary Fig. 12a)[7]. Using immunoblotting, we observed a significant reduction of ImpL2 protein in the hemolymph of *OE^{ts}>ImpL2^{RNAi}* flies (Supplementary Fig. 12e), confirming that oenocytes are indeed an important source of hemolymph ImpL2. Additionally, an increase of hemolymph ImpL2 protein could be found during starvation compared to fed conditions (Supplementary Fig. 12f, g).

Given the role of Desat1 in protein secretion, we tested whether secretion of ImpL2 was affected in *OE^{ts}>Desat1^{RNAi}* flies. For this, we overexpressed *ImpL2-HA* in oenocytes and conducted immunofluorescence using an HA antibody. Indeed, we found an increase of ImpL2-positive vesicles in cortical regions of oenocytes in *OE^{ts}>Desat1^{RNAi}* compared to *OE^{ts}>Control^{RNAi}* (Fig. 6f, g). As with the lipoproteins, we tested the effect of depolymerizing the excessive actin by overexpressing *tsr* or silencing *LIMK1*. Also here, these treatments led to a normalization of the ImpL2 levels in the hemolymph of *OE^{ts}>Desat1^{RNAi}* animals (Fig. 6h, i). Together, these results indicate that cortical actin depolymerization is able to rescue the defective secretion in *Desat1* KD oenocytes, thereby restoring proper ImpL2 secretion.

Next, we tested the role of ImpL2 in our NR assay using oenocyte-specific *ImpL2* KD and *ImpL2-HA* overexpression (OE). We found that the *ImpL2* KD group showed a significantly higher starvation sensitivity (Fig. 7a) and faster lipid turnover during starvation (Fig. 7b), whereas flies overexpressing *ImpL2* in oenocytes lived significantly longer and showed higher TAG than control flies during starvation (Fig. 7c, d). Accordingly, *ImpL2* OE decreased global *bmm* levels on day 3 of starvation, while the opposite could be seen for *ImpL2* KD (Fig. 7e). This was accompanied by an accumulation of LDs in the oenocytes (Fig. 7f, g), altogether suggesting that lack of ImpL2 secretion could lead to stimulation of FB lipolysis. We also found unchanged glycogen levels between fed and starved conditions in ImpL2 overexpression flies, while glycogen was reduced upon *Desat1* and *ImpL2* silencing during NR (Fig. 7h, i), suggesting that also carbohydrate stores are preserved by ImpL2 secretion from oenocytes.

As ImpL2 is a well-established insulin/insulin-growth factor signaling (IIS) antagonist[28,29] and as *bmm* is a target gene of Foxo that is inhibited by IIS[30], this raised the question in how far the IIS pathway is involved in the regulation of lipolysis. We observed that *ImpL2* OE induces nuclear Foxo in the FB and oenocytes, whereas Foxo remained outside the nuclei in *ImpL2* KD. While this confirms the antagonistic role of ImpL2 in IIS (Supplementary Fig. 13a), our snRNA-seq data showed *bmm* upregulation during starvation in FB cells but downregulation of *Thor* (4EBP in mammals), another Foxo target (Supplementary Fig. 13b). In addition, *Thor* was increased in oenocytes, arguing for cell-type specific IIS regulation during starvation. Accordingly, when silencing the *insulin receptor* (*InR*) in oenocytes, which also led to nuclear Foxo (Supplementary Fig. 13a), an enhanced survival was observed in our NR protocol, while the opposite was the case for *InR* overexpression (Fig. 7j). This correlated with increased and decreased TAG and glycogen levels during starvation, respectively (Fig. 7k, l). Interestingly, silencing and overexpressing *InR* in the FB showed similar but weaker effects. In particular, we could not detect a correlation between survival and nutrient stores as in the oenocytes (Supplementary Fig. 13c, d). Together, this suggests that IIS suppression by ImpL2, particularly in the oenocytes, is beneficial for prolonged starvation because it conserves energy resources.

## Discussion

Our study focuses on the role of oenocytes in regulating lipid dynamics in the hemolymph and the lipid storage functions of the FB during prolonged NR (Fig. 8). Specifically, we show 1) that prolonged NR leads

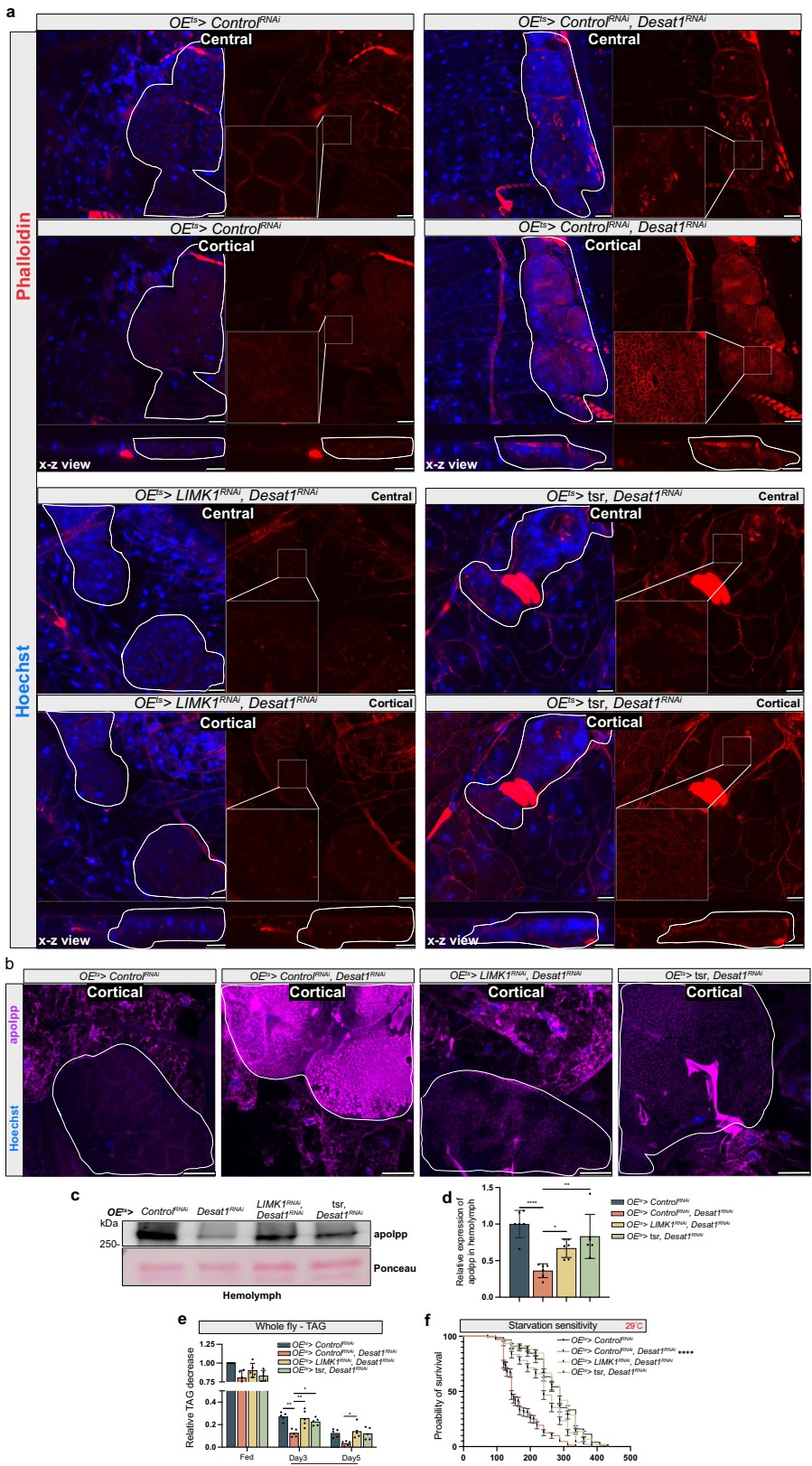

to changes in the circulating lipidome and to storage and release of lipids by oenocytes; 2) that oenocyte expression of *Desat1* is essential for driving the lipidomic changes and for promoting lipid storage in the neighboring FB; 3) that *Desat1* KD leads to lipoprotein sequestering on oenocyte surface and cortical actin accumulation; 4) that *Desat1* KD leads to impaired protein secretion, including ImpL2; 5) that cortical actin depolymerization rescues lipoprotein sequestration and ImpL2

secretion defects; and 6) that ImpL2 promotes survival resistance by controlling FB lipolysis.

As part of the starvation response, lipid dynamics are significantly altered during starvation, which is crucial for survival. Our results generally suggest that TAG storage in the FB is a central determinant of survival during fasting, which is consistent with previous reports[14,15]. However, as flies survived up to 10 days with our

**Fig. 5 | Lowering of F-actin in oenoncytes can rescue the sequestration of lipoproteins in $OE^{ts}$>$Desat1^{RNAi}$ animals. a** Representative immunofluorescence images of phalloidin (red) in central or cortical sections of oenocytes. Actin depolymerization via knocking down *Limk1* or overexpressing Tsr reduces cortical actin accumulation. $n = 4$, scale bars: 20 μm. **b** Representative immunofluorescence images of apoLpp in central or cortical sections of oenocytes. Dashed lines mark the oenocytes, $n = 3$, scale bars: 20 μm. **c** Western blot analysis of apolpp in the hemolymph. $OE^{ts}$>$Desat1^{RNAi}$ exhibit a reduced level of apolpp in the hemolymph compared with $OE^{ts}$>$Control^{RNAi}$, which can be rescued by actin depolymerization.

Only one band was seen after Ponceau staining, and this was used for normalization. **d** Quantification of apolpp in hemolymph, $n = 6$. Statistical tests: one-way ANOVA. *, $P < 0.05$; **, $P < 0.01$; ****, $P < 0.0001$. Data are presented as mean values ± SD. **e** TAG levels at different days of starvation in whole fly sample measured by TAG kit, $n = 5$. Statistical tests: Two-way ANOVA. **, $P < 0.01$; ***, $P < 0.001$. Data are presented as mean values ± SD. **f** starvation assay at 29 °C. $n = 150$, $P$ value was calculated using Log-rank (Mantel-Cox) test. $P < 1 \times 10^{-15}$ ($Control^{RNAi}$ vs. $Desat1^{RNAi}$). ****, $P < 0.0001$. Data are presented as percent survival values ± SEM. Source data for plots and exact $p$-value are provided as a source data file.

starvation medium, we were able to obtain additional insights into starvation-dependent lipid dynamics and tissue crosstalk. Over the course of a 7-day starvation period, TAG storage in FB and oenocytes was found to behave in an antiparallel fashion: while the first three days are dominated by massive FB lipolysis and oenocyte lipid uptake and storage, a reversal of functions can be observed thereafter. One consequence of the lipid exchange between these two tissues is that also the circulating lipidome is changing: we observed an increase in both the unsaturation level and carbon chain length of lipids in the *Drosophila* hemolymph during starvation, a finding that aligns with those from the human studies[1].

Why could a shift to unsaturated lipids be important during starvation? First, specific modified FAs released from oenocytes may function as lipokines, regulating metabolic homeostasis between distant organs. For example, Cao et al. reported on a lipid-mediated endocrine network where adipose tissue utilizes a monounsaturated fatty acid, C16:1n7-palmitoleate, as a lipokine in mice[31]. This suggests that similar mechanisms could be at play in other species, where modified fatty acids released by specific cells, such as oenocytes, act as signaling molecules to orchestrate systemic metabolic responses and maintain energy balance across various tissues. Second, unsaturated lipids generally enhance membrane fluidity, which could be essential for maintaining cell functionality under stress. Accordingly, an important requirement for Desat1 was previously demonstrated for the autophagy pathway in *Drosophila*[32]. And third, as proposed in this study, a more unsaturated circulating lipidome could also lead to more TAG storage in the FB. In line with the view that diglyceride acyltransferase 1 (DGAT1), the rate-limiting enzyme in TAG formation, prefers monounsaturated fatty acids over saturated ones[13], we could previously show that treatment with a SCD inhibitor leads to a significant reduction in LD formation in mouse kidney cells[12]. Here, we observe that *Desat1* KD in both oenocytes and FB cells causes a dramatic depletion of LD. In addition to these cell-autonomous effects, TAG storage in oenocytes was also shown to be a prerequisite for storage in the FB. Therefore, we hypothesize that Desat1-dependent cycling of lipids through the oenocyte TAG compartment will lead to an enrichment of unsaturated lipids in the circulation, promoting their storage in the FB over time.

The question how the exchange of lipids between FB and oenocytes may be carried out led us to take a closer look at the interaction between apolpp and Apoltp with the oenocytes. Historically, research into insect lipoproteins has suggested that these particles load and unload various lipids at the cell surface and function as reusable shuttles within the organism, facilitating the transport and distribution of lipids across different tissues[33–35]. Later, also endocytic mechanisms were found to be involved in the lipid exchange, at least in some tissues[17]. An intriguing example is the prothoracic gland where the KD of *Rab5* not only led to an accumulation of apolpp on the cell surface but also to impaired secretory functions, suggesting that in these cells the control of endo- and exocytosis is coupled via a lipid-dependent mechanism[36]. Our data suggest that oenocytes internalize FB-derived lipoproteins in a Rab5-dependent manner before releasing them again into the circulation. Upon *Desat1* KD, this release was impaired, with apolpp and Apoltp exhibiting a dramatic vesicular accumulation close to the surface. This accumulation correlated with impaired

endocytosis and an increased cortical actin polymerization[37], which most likely was the reason why actin depolymerization via cofilin/tsr overexpression or *LIMK* KD was able to improve the release of lipoproteins into the hemolymph as well as the overall survival of starved $OE^{ts}$>$Desat1^{RNAi}$ animals. However, more studies are needed to better understand the interplay of lipoproteins, endosomes and actin at the oenocyte cortex.

Beyond lipoproteins, we further identified ImpL2 as a factor that is upregulated in oenocytes during NR. Importantly, upon *Desat1* silencing, hemolymph ImpL2 levels dropped and overexpressed ImpL2 accumulated in the periphery of oenocytes, suggesting that an impaired release of ImpL2 in $OE^{ts}$>$Desat1$ KD animals is at least partially responsible for the starvation sensitivity of these animals. While also here actin depolymerization led to an increased ImpL2 release, it is unclear whether the cortex alterations in Desat1-deficient oenocytes or other actin filament functions along the secretory pathway are responsible for the secretion defect of ImpL2. In islet cells under low glucose conditions, for example, treatment with an actin depolymerization agent can enhance insulin secretion by up to 20 times, highlighting the impact of cytoskeletal dynamics on cellular secretion mechanisms[22].

Our data show that ImpL2 secreted from oenocytes can influence FB lipolysis. *ImpL2* overexpression in oenocytes was shown to suppress *bmm* expression, and this led to more TAG stores and better survival during starvation, while the opposite was true for *ImpL2* KD. At first glance, it appears counter-intuitive that an inhibitor of insulin promotes starvation resistance given that TAG storage is usually promoted by insulin[38]. On the other hand, insulin signaling also promotes anabolic energy-consuming processes like cell growth, which could be detrimental during fasting and aging[39,40]. Accordingly, some dILPs, such as dILP6 and dILP7, increase during starvation in *Drosophila*[41], and flies with genetic ablation of dILP-producing IPCs show increased TAG levels and higher resistance of starvation[42]. Likewise, flies with mutations in *chico*, a key component of the insulin pathway, exhibit increased lipid levels despite a reduced body size[43]. Furthermore, previous studies could show that larvae with a loss-of-function mutation in *ImpL2* exhibit increased sensitivity to NR conditions[29] and that ImpL2 contributes to energy conservation in an IIS-independent manner during the transition from larvae to pupae[44]. It is therefore conceivable that both inter-organ and local insulin signaling needs to be tightly controlled in specific dietary and developmental conditions, and that in prolonged starvation the secretion of ImpL2 from oenocytes contributes to this balance.

Overall, our study contributes to the understanding of lipid metabolism during the starvation response. By manipulating Desat1 function, we demonstrate a key role of oenocytes in modifying circulating lipid profiles as well as FB function. As a medical implication, the cell-autonomous and non-cell-autonomous effects observed upon *Desat1* KD need to be considered when testing SCD inhibitors in clinical trial[45,46]. Moreover, serum levels of the ImpL2 homolog in humans, the insulin growth factor binding protein 7 (IGFBP7), have been shown to be positively correlated with insulin resistance, body mass index and the risk of metabolic syndrome[47,48], while the treatment of mouse and human hepatocytes with IGFBP7 increased LDL and VLDL production. Therefore, ImpL2/IGFBP7 may not only represent an ancient hormone

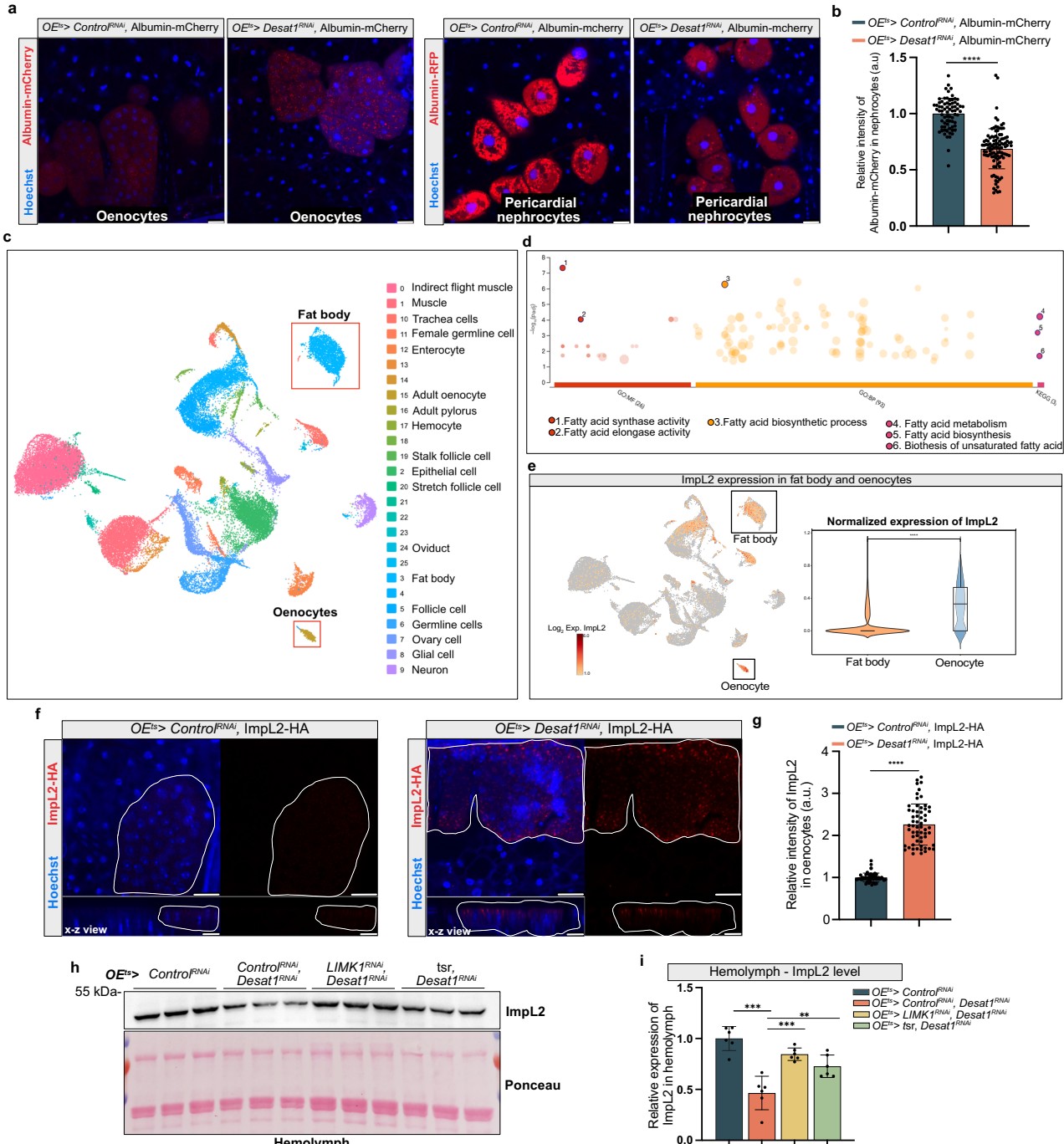

**Fig. 6 | Single-nucleus RNA-seq identifies *ImpL2* as important factor during starvation. a** Representative immunofluorescence images of albumin-mCherry in oenocytes or pericardial nephrocytes. Albumin-mCherry secreted by oenocytes into the hemolymph is taken up by pericardial nephrocytes. *n* = 4. scale bars: 20 μm. **b** Quantification of relative intensity of albumin-mCherry in pericardial nephrocytes, each data point represents the relative intensity of albumin-mCherry in a pericardial nephrocyte. scale bars: 20 μm. $P < 1 \times 10^{-15}$ (*OE^ts^> Control^RNAi^*, Albumin-mCherry vs. *OE^ts^> Desat1^RNAi^*, Albumin-mCherry). Statistical tests: two-tailed unpaired t test. ****: *P* < 0.0001, each data point presents a pericardial nephrocytes, *n* = 4. Data are presented as mean values ± SD. **c** A uniform manifold approximation and projection (UMAP) plot. Each color and dot in the plot represented a cluster and a single nucleus, respectively. **d** Gene enrichment analysis of these significant different genes (day 2 of starvation vs. fed) was performed demonstrating that many genes were involved in lipid metabolism. **e** *ImpL2*

expression pattern shows a high expression level of *impL2* in oenocytes. ****: *P* < 0.0001. **f** ImpL2-HA overexpression in oenocytes with or without *Desat1* deficiency. ImpL2-HA (red) accumulates in oenocytes in the *Desat1* KD group. X-z view shows cortical accumulation in oenocytes (marked by dashed line), *n* > 4, scale bars: 20 μm. **g** Quantification of relative intensity of ImpL2-HA in oenocytes, each data point represented the relative intensity of ImpL2-HA signal in a cluster of oenocytes, *n* = 4, data are presented as mean values ± SD. $P < 1 \times 10^{-15}$. Statistical tests: multiple two-tailed unpaired *t* tests. **h** Western blot analysis of ImpL2 in the hemolymph. *OE^ts^>Desat1^RNAi^* exhibited a reduced level of ImpL2 in the hemolymph, which can be rescued by actin depolymerization, *n* = 5. **i** Quantification of ImpL2 level in hemolymph. Statistical tests: one-way ANOVA, **, *P* < 0.01; ***, *P* < 0.001, each data point represented a biological independent hemolymph sample from around 30 adult flies, *n* = 5. Data are presented as mean values ± SD. Source data for plots and exact p-value are provided as a source data file.

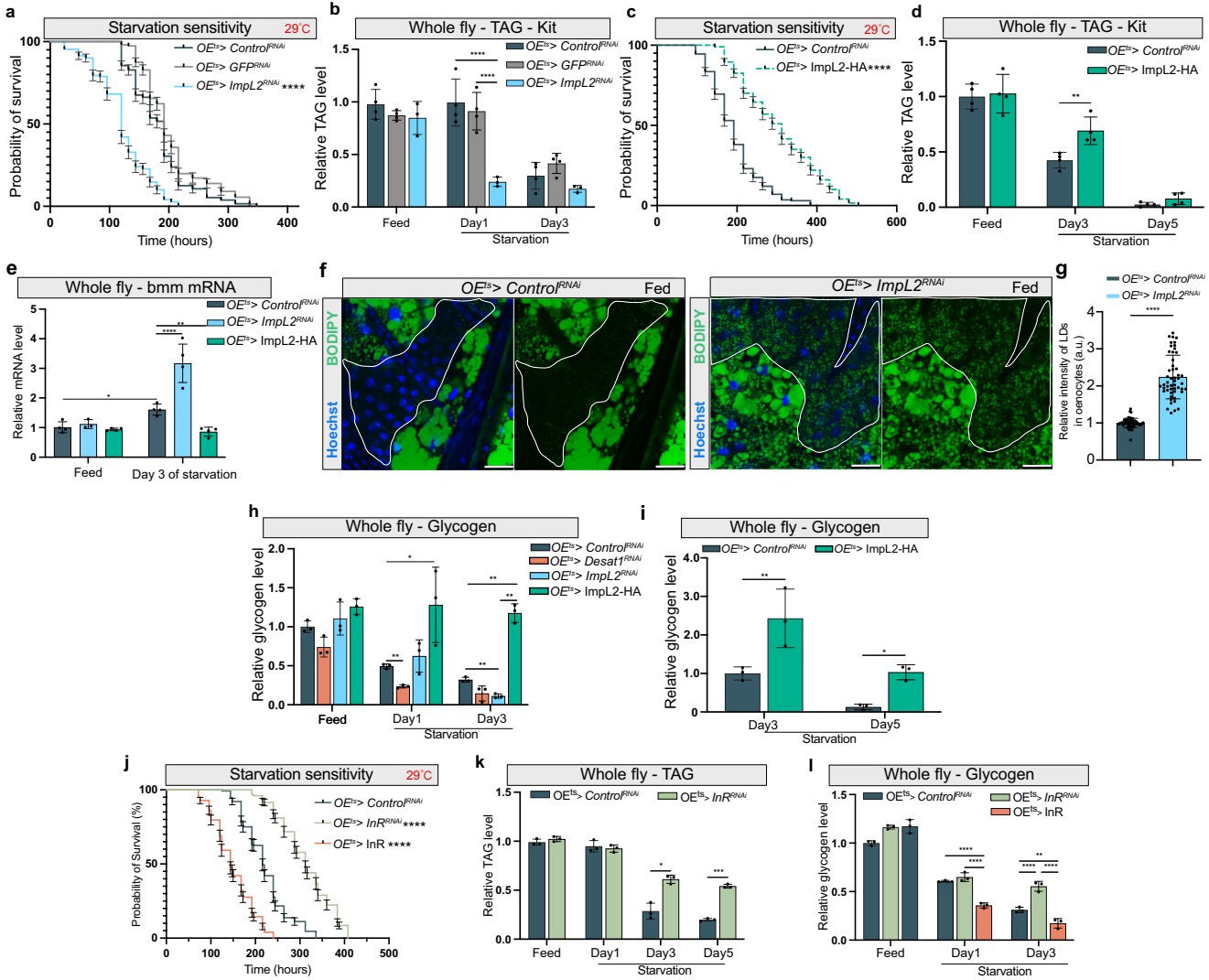

**Fig. 7 | ImpL2 secreted from oenocytes preserves TAG levels and increases starvation resistance. a** Starvation assay in flies with oenocyte-specific *ImpL2* knockdown at 29 °C. Both *OE^ts^>Control^RNAi^* and *OE^ts^ > GFP^RNAi^* flies were represented control groups. *n* = 151. Data are presented as mean values ± SEM. *P* value was calculated using Log-rank (Mantel-Cox) test. ****, *P* < 1×10⁻¹⁵, *OE^ts^>ImpL2^RNAi^* vs. *OE^ts^>Control^RNAi^* or *OE^ts^ > GFP^RNAi^*. **b** TAG levels in whole fly sample were measured at different time point of starvation. *n* = 3. Data are presented as mean values ± SD. Statistical tests: two-way ANOVA with Tukey's multiple comparisons test. ****, *P* < 0.0001. **c** Starvation assay in flies with oenocyte-specific *ImpL2-HA* over-expression at 29 °C. *n* = 200. Data are presented as mean values ± SEM. *P* value (*P* < 1×10⁻¹⁵) was calculated using Log-rank (Mantel-Cox) test. ****, *P* < 0.0001. **d** TAG level in whole fly samples was measured by TAG kit at different time point of starvation. *n* = 4. Data are presented as mean values ± SD. **: *P* < 0.001(Day3, *Control^RNAi^* vs. ImpL2-HA). Statistical tests: two-way ANOVA. **e** mRNA level of bmm in whole fly samples during starvation. Statistical tests: two-way ANOVA with Tukey's multiple comparisons test. *, *P* < 0.05; **<0.01; ****, *P* < 0.0001. *N* = 3. Data are presented as mean values ± SD. **f** LDs (green) during fed condition in oenocytes with

*ImpL2* knockdown, *n* = 3. Dashed lines mark the oenocytes. scale bars: 20 μm. **g** Quantification of relative intensity of LDs in oenocytes, each data point represented the relative intensity of BODIPY signal in a cluster of oenocytes. Statistical tests: two-tailed unpaired t test, ****, *P* < 1×10⁻¹⁵ (*Control^RNAi^* vs. *ImpL2^RNAi^*). Error bars indicate standard deviation. **h**, **i** Glycogen level of flies during early stage or later stage of starvation measured with whole fly samples. Statistical tests: two-way ANOVA. *, *P* < 0.05; **, *p* < 0.01, *n* = 3. Data are presented as mean values ± SD. **j** Starvation sensitivity in flies with *InR* KD or *InR* OE specific in oenocytes. *n* = 156. Data are presented as mean values ± SEM.1 P value was calculated using Log-rank (Mantel-Cox) test. ****, *P* < 1 × 10⁻¹⁵ (*Control^RNAi^* vs. *InR^RNAi^* or InR group). **k** TAG level of flies with oenocytes specific InR KD at different days of starvation. *n* = 3. Data are presented as mean values ± SD. Statistical tests: two-way ANOVA. *, *P* < 0.05; ***, *P* < 0.001. **l** Glycogen level in flies with oenocyte-specific *InR* KD or *InR* OE at different days of starvation. *n* = 3. Data are presented as mean values ± SD. Statistical tests: two-way ANOVA, **, *P* < 0.01; ****, *P* < 0.0001. Source data for plots and exact *p*-value are provided as a source data file.

that promotes fat storage in the adipose tissue but also a suitable drug target for the treatment of metabolic syndrome in humans.

## Methods

### *Drosophila* husbandry

Flies were reared at either 25 °C or 18 °C with 65% humidity under a 12-h light/dark cycle, fed standard laboratory food. For experiments conducted at 29 °C, such as the starvation survival assay, crosses were initially maintained at 18 °C until they reached the adult stage. Two-

day-old adult flies were then collected and transferred to an environment at 29 °C and 65% humidity. These conditions were maintained for 1 week with a 12-h light/dark cycle, during which flies were fed either standard or modified starvation food before beginning the experiments (Supplementary Table 4).

### Fly stocks

Flies used in this study are listed in Supplementary Table 1. For the Desat1 KD we used two independent RNAi lines (V104350KK and

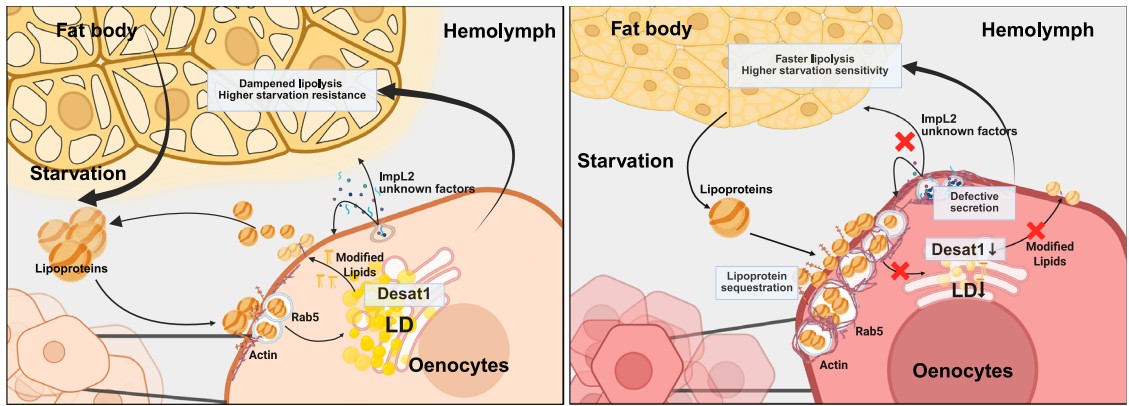

**Fig. 8 | Overview of the role of oenocytes in systemic metabolism during starvation and the observed phenotypes upon Desat1 deficiency.** Schematic created in BioRender. Li, J. (2025) (https://BioRender.com/7uffvc1, https://BioRender.com/wgm1ciq) is licensed under CC BY 4.0.

v33338GD from VDRC, data shown here were obtained with the GD line). UAS-albumin-mCherry flies were generated by subcloning human albumin from the pGEM-T albumin vector (Sino Biologicals #HG10968-G) first into pmCherry-N1 (Clontech) before subcloning albumin-mCherry further into pUASattB. The construct was injected into flies with the attP landing site at 51C1 by Bestgene, Inc.

Lexaop shRNAs (21 bp) were cloned into pLexAop-WALIUM20 vector digested with EcoRI+XbaI, as described previously[49,50]. The oligos were included in Supplementary Table 3.

For phiC31-integration, each plasmid was injected at 50 ng/µl into y w nos-phiC31-int; attP2. Injected male G0 flies were crossed with y w; MKRS/TM6B to identify transformants and remove the integrase from the X chromosome, and subsequently balanced. All transgenic lines were sequenced to confirm the identity of the shRNA.

### Conditional expression of UAS transgenes
The oenocyte-specific driver, *PromE-Gal4*[51], was combined with a ubiquitously expressed temperature-sensitive Gal80 (OE^ts >). To activate the PromE-gal80^ts driver, flies were first maintained at 18 °C until the adult stage, and two days after eclosion, flies were transferred to 29 °C for 1 week.

### Starvation sensitivity assay in adult flies
To generate age-synchronized adult flies, larvae were raised on standard laboratory food at a low density at 18 °C. After eclosion, 2-day-old flies were transferred to conditions of 29 °C and 65% humidity and allowed to mate for 1 week. The flies were anesthetized using a low level of $CO_2$, and mated females were sorted at a density of 15 per vial. Flies were transferred to fresh food vials every second day, and mortality was monitored twice daily. The modified starvation food medium was prepared according to the protocol described[19] with modifications. This medium contains all physiologically relevant ions, including biometals, but lacks energy sources such as sugars, proteins, amino acids, lipids, lipid-related metabolites, nucleic acids, and vitamins (Supplementary TableS 5 and 6).

### Cold exposure experiment
Flies were reared at 18 °C and 2-day-old flies were transferred to conditions of 29 °C and 65% humidity and allowed to mate for 1 week. After that, male flies were transferred to a fresh vial (15 flies per vials, around 15 vials for each group) and then kept for 24 h at 4 °C before transferring them to 25 °C for flies for recovery. After 2 h, the number of recovered flies were recorded. Female flies were kept for 48 h at 4 °C.

### Immunostaining, lipid droplet labeling, and tissue imaging
Dorsal abdominal cuticles containing both FB and oenocytes were dissected in PBS and fixed with 4% paraformaldehyde for 25 min at room temperature. After fixation, the cuticles were washed for three times and incubated with 0.1% PBST (PBS with 0.1% Triton x-100) for 30 min. Then, samples were incubated with a solution of 5% goat serum for 30 min blocking at room temperature before incubating overnight with primary antibodies at 4 °C diluted in 0.1% Tween 20 (guinea pig anti-apolpp[52], 1:500; rabbit anti-Apoltp, 1:1000 (both from former Eaton lab)[16]; mouse anti-HA, 1:200, (from Santa Cruz sc-7392); rabbit anti-Rab5, 1:200 (from Abcam ab31261); anti-foxo, 1:200 (from R.Tjian). Following this step, the cuticles were washed three times for 10 min each and incubated with fluorescent conjugated secondary antibodies (Alexa Fluor 488, Alexa Fluor 555, Alexa Fluor 647 from Invitrogen) at a dilution of 1:200, Alexa Fluor™ 555 Phalloidin at dilution of 1:500 and Hoechst at 1:1000 in 0.1% Tween 20 for 2 h at room temperature. For lipid droplet labeling, BODIPY 493/503 (D3922, Thermo Scientific) was diluted to 2.5 µg/ml in PBS and added during the Hoechst staining. Tissues were mounted in Antifade mounting medium (H-1000, Biozol). Images were performed on either a Leica SP5 or Zeiss LSM 780 confocal microscope using a 63× oil-immersion objective. Images were processed using Fiji software.

### Quantitative PCR (QPCR)
Total RNA was extracted using the Quick-RNA Tissue/Insect Kit (Zymo Research) according to the manufacturer's instructions (5 whole flies, 10 cuticles, or 20 heads per sample). For cuticle samples, dissections were performed on ice and the gut, heart, and reproductive tissues were removed to minimize contamination. RNA concentration and purity were measured with a NanoDrop spectrophotometer. For cDNA synthesis, 500 ng of total RNA per sample was reverse transcribed using PrimeScript™ RT Master Mix following the manufacturer's protocol. Quantitative PCR (qPCR) was performed with a SYBR Green master mix (PCR biosystems) on a QuantStudio 3 Real-Time PCR System (Thermo Fisher Scientific). Primer sequences are listed in Supplementary Table 2.

### Glycogen measurement
Glycogen levels were quantified using the Glycogen Assay Kit (MAK465, Sigma-Aldrich) following the manufacturer's instructions. For each biological replicate, five adult flies were pooled (three to four biological replicates per condition). Samples were homogenized on ice in 500 µL lysis buffer (2.5 g/L NaF, 25 mM citric acid, pH 4.2) and centrifuged at 14,000 $g$ for 5 min at 4 °C. Supernatants were transferred to

a clear, flat-bottom 96-well plate. After addition of the kit working reagent(s) and incubation at room temperature for 30 min, absorbance at 570 nm was measured using a Spark multimode microplate reader (Tecan).

## Hemolymph extraction and Western blots

To extract hemolymph from adult flies, 30–35 flies were first anesthetized using $CO_2$. Each fly was then carefully pierced at the thorax with a tungsten needle and immediately transferred to 0.5 ml Eppendorf tubes that had three holes punctured at the bottom using a 24-gauge needle. These tubes were then placed into a 1.5 ml Eppendorf tube and kept on ice. The samples were centrifuged at 2100 g for 5 min at 4 °C. Following centrifugation, 1 µl of hemolymph was transferred to a new Eppendorf tube and snap-frozen. For Western blot analysis, 1 µl of hemolymph was lysed in RIPA buffer (Thermo Scientific, #89900) supplemented with a protease inhibitor cocktail (Roche, 11697498001) and a phosphatase inhibitor (Roche, #04906845001). The lysate was shaken properly at 4 °C for 30 min. Subsequently, 4x Laemmli sample buffer (Bio-Rad) and 2% beta-mercaptoethanol were added to the samples, which were then boiled for 5 min at 95 °C. Lysates were subjected to SDS-PAGE electrophoresis using either 4–15% Mini-PRO-TEAN® TGX™ Precast Protein Gels from Bio-Rad or homemade SDS gels. Proteins were then transferred onto nitrocellulose membranes using the iBlot2 dry blotting system (iBlot™ 2 Transfer Stacks, Thermo Fisher). After the transfer, the membrane was stained with Ponceau S for 10 min to verify protein transfer and then blocked in Tris-buffered saline with 0.1% Tween-20 (TBST) containing 5% milk at room temperature for 90 min. The membrane was incubated with the primary antibody overnight at 4 °C (rat anti-apolpp, 1:3000 (from former Eaton lab)[52]; rabbit anti-ImpL2, 1:2000 (from L.Partridge)[53]. Subsequent washes involved three 10-min TBST rinses before incubation with a horseradish peroxidase (HRP)-conjugated secondary antibody (anti-rat, 1:10,000, 31470, Thermo Scientific; anti-Rabbit, 1:10,000, SA1-200, Thermo Scientific) in TBST for 90 min at room temperature. After four additional 10-min washes in TBST, protein bands were visualized using SuperSignal West Dura Extended Duration Substrate (34076, Thermo Scientific).

## TAG measurement

We utilized the Triglyceride Assay Kit (TR0100) from Sigma to measure TAG levels. For the assay, three flies constituted one replicate, with three replicates analyzed in total. Lipids were extracted following the kit's protocol. Each sample, consisting of three flies and 300 µl of 5% NP-40/ddH2O, was added to a 2 ml tube containing a tungsten carbide bead (Qiagen, 69997). The samples were homogenized using a pre-cooled TissueLyser LT for 3-4 cycles, each lasting 4 min. Samples were then heated to 90 °C in a water bath for 4 min, cooled to room temperature, before repeating the heating process once more. After two heating cycles, samples were centrifuged for 2 min at maximum speed in a microcentrifuge. The supernatant was diluted fivefold with NP-40/ddH2O before proceeding with the assay. TAG concentrations were measured using a Spark® Multimode Microplate Reader at OD 540 nm and calculated against a standard curve.

## Lipidomics

Mass spectrometry-based lipid analysis was conducted by Lipotype GmbH (Dresden, Germany), using methods described by ref. 53. Lipids were extracted following the two-step chloroform/methanol procedure outlined by ref. 54. The samples were spiked with an internal lipid standard mixture containing various lipid markers (e.g., CL, Cer, DAG, HexCer, LPA, LPC, LPE, LPG, LPI, LPS, PhA, PC, PE, PG, PI, PS, CE, SM, Sulf, TAG, and Chol). Post-extraction, the organic phase was transferred to an infusion plate and dried using a speed vacuum concentrator. The first-step dry extract was resuspended in 7.5 mM

ammonium acetate in a chloroform/methanol/propanol solution (1:2:4, V:V:V), and the second-step dry extract in a 33% ethanol solution of methylamine in chloroform/methanol (0.003:5:1; V:V:V). Liquid handling was performed using a Hamilton Robotics STARlet robotic platform, equipped with anti-droplet control for precise organic solvent pipetting. Samples were analyzed by direct infusion on a QExactive mass spectrometer (Thermo Scientific) equipped with a TriVersa NanoMate ion source (Advion Biosciences). Analyses were conducted in both positive and negative ion modes, with a resolution of Rm/z = 200 = 280,000 for MS and Rm/z = 200 = 17,500 for MSMS, in a single acquisition. MSMS triggers were set by an inclusion list covering the MS mass ranges in 1 Da increments[55]. MS and MSMS data were integrated to monitor various lipid ions: CE, DAG, and TAG as ammonium adducts; PC and PC O- as acetate adducts; and CL, PA, PE, PE O-, PG, PI, and PS as deprotonated anions. MS only was used to track LPA, LPE, LPE O-, LPI, and LPS as deprotonated anions; Cer, HexCer, SM, LPC, and LPC O- as acetate adducts; and cholesterol as an ammonium adduct of an acetylated derivative[56]. Data were processed and analyzed using in-house-developed lipid identification software, LipidXplorer, and managed with an in-house data system. Only lipid identifications with a signal-to-noise ratio >5 and signal intensity at least fivefold higher than corresponding blanks were considered for further analysis. Results were visualized using Prism9 software.

## Single-nucleus RNA sequencing

Single nuclei suspensions were prepared following the protocol described in the Fly Cell Atlas[27]. For the preparation, whole-body flies, excluding heads, were flash-frozen in liquid nitrogen and subsequently homogenized in 1 ml of Dounce buffer. The buffer composition included 250 mM sucrose, 10 mM Tris (pH 8.0), 25 mM KCl, 5 mM MgCl_2, 0.1% Triton-X, 0.5% RNasin Plus (Promega, N2615), 1X protease inhibitor (Promega, G652A), and 0.1 mM DTT. The homogenate was then filtered through a 40 µm cell strainer and a 40 µm Flowmi cell strainer (BelArt, H13680–0040) to obtain the suspension. After initial preparation, the samples were centrifuged, washed, and resuspended in 1X PBS containing 0.5% BSA and 0.5% RNasin Plus. The suspension was then filtered through a 40 µm Flowmi cell strainer (BelArt, H13680–0040) immediately before fluorescence-activated cell sorting. For nuclear staining, DRAQ7™ Dye (Invitrogen, D15106) was used. Sorting was performed using a Sony SH800Z Cell Sorter at the Systems Biology Flow Cytometry Facility at Harvard Medical School. Post-sorting, the nuclei were collected and resuspended at a concentration of 700–800 cells/µl in 1X PBS with 0.5% BSA and 0.5% RNasin Plus.

Single-nucleus RNA sequencing (snRNAseq) was performed following the 10X Genomics protocol (Chromium Next GEM Single Cell 3' v3.1 Rev D). We conducted two reactions for flies in fed and two reactions for flies in starvation condition at each time point (day 2 and day 5). The snRNAseq data were processed using the Cellranger count pipeline 6.1.1 to generate feature-barcode matrices. To normalize data across different samples, the read depths were equalized, and the matrices were aggregated into a single feature-barcode matrix using the Cellranger aggr pipeline. In total, 43,562 cells were profiled, including 20,395 fly cells in fed condition and 15,264 fly cells at Day 2 of starvation, and 7903 control fly cells at Day 5 of starvation. Cell clusters and gene expression levels were visualized using the Loupe Browser 6.

## Fluorescence recovery after photobleaching (FRAP)

The FRAP experiment was conducted using a Zeiss 780 m confocal microscope equipped with a Plan-APOCHROMAT 63×/1.4 Oil objective. Fresh abdominal tissue was dissected in PBS and placed on a glass slide with a thin layer of Halocarbon 700 oil (H8898, Sigma) spread on it. A 40 mm × 60 mm coverslip was then carefully

positioned over the tissue. After allowing the setup to stabilize for 6–10 min, the slide was mounted on the microscope for imaging and the FRAP experiment. The same settings were maintained for all FRAP experiments: Frame size was set at 512 × 512 pixels with a zoom of 2. Gain settings varied between 600 and 850, adjusted for each slide to maximize signal without causing overexposure. Time series were recorded at 3-s intervals, with the duration ranging from 2 to 5 min, depending on the observed recovery time. The photobleaching region size was set at 30, and the pinhole was adjusted to 1 Airy Unit (AU). Using Zeiss ZEN 2010 software, the following regions of interest (ROI) were extracted: bleach (BL): the main ROI bleached by the laser; background (BG): a region of only noise (outside of any target fluorescence); reference (REF): a region of fluorescence outside of the bleached region. This is used to determine bleaching as a result of repeated image acquisition. The normalized data was calculated: $BL\_corr2(t) = BL\_corr1(t)/REF\_corr1(t) = [BL(t) - BG(t)]/[REF(t)\cdot BG(t)]$ and processed by Prism 9.

## Statistical analysis

Statistical analysis for lifespan was performed by Log-Rank test. Full details of survival data are provided in corresponding figure legend and source data. Other data were analyzed by Student's $t$ test or ANOVA as appropriate in GraphPad Prism v8–10.

## Reporting summary

Further information on research design is available in the Nature Portfolio Reporting Summary linked to this article.

## Data availability

Raw snRNA-seq reads and gene2cell matrix file have been deposited in the NCBI Gene Expression Omnibus (GEO) database under accession code GSE284161. Processed datasets can be mined through a web tool [https://www.flyrnai.org/scRNA/] that allows users to explore genes and cell types of interest. The other data generated in this study are provided within the manuscript and its Supplementary Information, or from the corresponding author upon reasonable request. Source data are provided with this paper.

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

## Acknowledgements

We thank Zvonimir Marelja for generating UAS-albumin-mCherry flies. We also want to thank Jonathan Zirin for design of the LexAOP-Bmm RNAi constructs, Lu-Ping Liu and Corey Forman for generation of the Bmm fly lines. We are grateful to Claire Hu and Weihang Chen from DRSC/TRiP Functional Genomics Resources & DRSC-BTRR. We thank the former Eaton lab, Robert Tjian and Lina Partridge for providing antibodies as well as Mike Henne, Seung Kim, Samuel Liegeois, Ronald Kühnlein, Yohanns Bellaiche, Bruno Lemaitre, the Bloomington stock center, FlyORF and Vienna *Drosophila* RNAi Center (VDRC) for providing flies. We thank Wilhelm Palm for critical reading of the manuscript. We acknowledge the European Research Council (ERC) under the European Horizon 2020 research and innovation program (Grant agreement No. 865408 (RENOPROTECT), the Deutsche Forschungsgemeinschaft (DFG SI1303/5-1 (Heisenberg-Programm) (all to M.S.). Work in the Perrimon Lab is supported by NIH/NCI Grant #5P01CA120964-15 and is delivered as part of the CANCAN team supported by the Cancer Grand Challenges partnership funded by Cancer Research UK (CGCATF-2021/100022) and the National Cancer Institute (1 OT2 CA278685-01). Y.L. is supported by the Sigrid Jusélius Foundation (Sigrid Juséliuksen Säätiö), Finnish Cultural Foundation (Suomen Kulttuurirahasto). N.P. is an investigator of the Howard Hughes Medical Institute.

## Author contributions

Conceptualization: J.L., M.S.; Methodology: J.L., K.H., Y.L; Investigation: J.L., I.D., K.H., Y.L.; Resources: N.P., M.S.; Writing—Original draft: J.L., M.S.; Writing—Review and editing: J.L., K.H., Y.L., N.P., M.S.; Visualization: J.L.; Supervision: N.P., M.S.; Funding acquisition: N.P.; M.S.

## Funding

## Competing interests

The authors declare no competing interests.
