## [Transparent Peer Review file · Nature Communications]

Desaturase-dependent secretory functions of hepatocyte-like cells control systemic lipid metabolism during starvation in *Drosophila*

Corresponding Author: Professor Matias Simons

Version 0:

Reviewer comments:

Reviewer #1

(Remarks to the Author)

In the manuscript „Desaturase-dependent secretory functions of hepatocyte-like cells control systemic lipid metabolism during starvation in *Drosophila*“, Li and colleagues explore the role of Desat1 in oenocytes for systemic regulation of lipid metabolism and lipolysis during starvation in larval and adult *Drosophila*.

The study provides new insights on the role of hepatocyte-like oenocytes in regulating energy distribution from lipid storage in the fat body during starvation. Here the authors give detailed mechanistic insights on the role of Desat1 in oenocytes as a key regulator of lipid metabolism by regulating soluble factors and their exocytosis, such as Impl2, but also by altering lipoprotein sequestration and endocytosis in oenocytes. Overall, the authors provide profound evidence that oenocytes are key regulators of energy mobilization and inter-organ communication with the fat body to modulate systemic lipid metabolism during starvation.

The manuscript and findings are of high relevance to the field, whereas oenocytes are often neglected or underexplored in their role during stress responses and infection, where the organism has high needs in rapidly adjusting energy storage and mobilization.

The manuscript is well written and I really enjoyed reading it. While reading, a couple of questions came up which the authors could additionally test to extend their study and make their findings even more solid before publication.

- 1) In the beginning, when the authors describe their new NR model, it is unclear which fly line is used, please clarify. Furthermore, I think it is important to show the overall survival of the flies on NR in Figure 1a and Ext. Figure 1a.
- 2) Temperature plays critically into the regulation of metabolic activity in flies. If possible, I recommend to show NR wildtype survivals and the induced changes in lipid metabolism at 25C and in the same direction, it would be good to show the effects of Desat KD maybe with an auxin inducible Gal4 approach in parallel.
- 3) One of the most interesting findings is that the data provided here highlights again the essential temporal regulation of lipid metabolism in the fat body. As the authors pointed out, before lipolysis is initiated in the fat body and the oenocytes start sequestration of lipoproteins (but also in parallel), systemic carbohydrate metabolism and insulin signaling is extremely adjusted and important for efficient energy mobilization. On nutrient restriction it would be helpful if the authors could provide additional insights on the regulation of carbohydrates and insulin signaling (activity on protein level (pAKT and foxo TF translocation and /or regulation of foxo targets) at different stages of NR and Desat KD in oenocytes.
- 4) Impl2 is a negative regulator of insulin signaling therefore it would be interesting to understand the regulation of insulin signaling by Impl2 is seen in the fat body and affects
- 5) Is the decreased release of Ilp2, 3, 5 from IPCs in the brain affected by Desat1 KD or overexpression of Impl2 during NR (more a control experiment to prove that this is not the case)?
- 6) The authors suggest lipid flux between oenocytes and fat body during NR. I would suggest here to do ¹³C-FFA feeding before NR onset to monitor the precise flux of lipids and also to better determine the dynamics seen in the lipidomics and to provide the final proof that lipids are fluxing from the fat body to oenocytes and back in distinct lipid species.
- 7) The time-resolved lipidomics of hemolymph and whole fly are very interesting. However, I think it would be interesting to answer how the lipid composition in the fat body (and oenocytes) are changing. Whereas oenocytes might be tricky in lipidomics, but analysis of dorsal abdominal fat body could give here more insights.

- 8) At some point in the manuscript, the authors state that the Desat1 in oenocytes is important for lipid storage in the neighbouring fat body. Here, the authors only provide evidence for this in late stages of NR, so I would be specific that this is during NR and not steady state or test if Desat1 in steady state might also affect TAG storage and LD formation in fat body. Furthermore, I am not convinced yet that this is only specific to neighboring fat body, the authors should test in adult *Drosophila* if Desat1 in oenocytes will also affect distant fat body for example in the head behind the optical lobes.
- 9) Can the authors elaborate a bit further on the gut during NR and lipid dynamics here. Are there any effects during NR compared to “standard” starvation? Does Desat1 KD or Impl2 OE also affect the gut, especially in terms of lipoprotein sequestration and lipolysis in this organ at early stages of starvation.
- 10) The authors state in their manuscript: “Together with the strong LD decline in the FB of OETs> Desat1RNAi animals, this indicates that oenocytes non-autonomously promote lipid storage in the FB, presumably by suppressing lipolytic activity”. From my point of view this has to be either stated specifically for starvation/NR or it has to be shown that this is also the case during steady state (I am not sure if I can see or interpret that from the data provide in the fed stage with Desat1 KD)
- 11) Is Impl2 the only adipokine or factor that is affected by changes in exocytosis upon Desat1 KD in oenocytes?
- 12) Is ILP6 signaling from the fat body to oenocytes affected by Desat1 KD in oenocytes?
- 13) The effects of Desat1 KD in fat body are interesting and showing profound changes during development, however somehow do not support the storyline of the manuscript. The authors should consider inducible Desat1 KD in fat body during NR in adult flies similar to the experiments with oenocyte-specific KD.
- 14) Minor point: Figures should appear in numeric order in the text. Figure 1g is missing the labeling where the higher magnification is taken from.
- 15) Minor point: In Figure 6g it looks like there are only two samples in a group at day 1, but statistics were applied, please remove the stats or add samples to the analysis.
- 16) For all bar graphs: please indicate individual samples with dots on top of the bars, as done in Figure 5 and 6.
- 17) Please provide a scale bar for the higher magnification inserts in some of the imaging figure panels.

Reviewer #2

(Remarks to the Author)

This study investigates the roles of *Drosophila* liver-like oenocytes in lipid metabolism and interorgan lipid transport. It begins with a lifespan analysis and lipid analysis of flies fed an ion-enriched starvation diet, showing a lifespan extension and changes in lipid storage, particularly in females. Lipid changes include an increase in double bond lipids, suggesting an increase in lipid desaturation. They find that depletion of the *Drosophila* fatty acid desaturase Desat1 specifically in oenocytes leads to changes in the lipid profile of the blood and reduced fat storage in the adipose-like fat body. Desat1 depletion also causes more starvation sensitivity. There were also changes in oenocyte cortical actin and in lipoprotein secretion into the blood. Insulin influencing protein Impl2 (mammalian IGFBP7) secretion from oenocytes is also reduced with Desat1 knockdown.

This work is important and nicely dissects how oenocytes significantly impact the lipid profile of secreted lipids in the animal hemolymph. It also investigates the relationship between lipid starvation profile and aging, as well as cold tolerance and other stresses like nutrient restriction. The analysis on lipid transport and the actin cytoskeleton is also intriguing, although additional work is needed to provide mechanistic insights. In general the study is well executed, although some conclusions need additional data to be supported by the observations. A general concern is that the changes observed with Desat1 depletion in oenocytes could result in general ER stress or other cellular stresses, and thereby phenotypes may represent pleiotropic effects.

Concerns:

- 1) A general concern is the Desat1 RNAi depletion in the oenocytes can lead to many changes in these cells in addition to lipid saturation changes. More saturated lipids can lead to ER stress and perturbed organelle homeostasis for many organelles including mitochondria. Determining whether ER stress is elevated in oenocytes in Desat1 RNAi, and whether mitochondrial function is perturbed, are important controls that provide context for this investigation.
- 2) Desat1 RNAi in oenocytes resulted in reduced LDs in the FB over time. This is proposed as due to an increase in lipolysis in the FB (Fig 3e), but lipid biosynthesis may also be altered. Can lipolysis in the FB be investigated further? Does Bmm loss in the FB in this condition rescue the LD levels? A heterozygous *bmm*(1) mutant is utilized in Fig3g,h, but additional analysis is required to make this conclusion.
- 3) Desat1 RNAi in oenocytes led to increased Apolpp lipoprotein accumulation at the oenocyte cell surface together with cortical actin. This is a key new observation and suggests actin dynamics influence lipoprotein lipid delivery at oenocytes. How does directly modulating oenocyte actin pools by depletion of other actin components of associated proteins impact lipid delivery to these cells? The implication is that actin promotes endocytic uptake into Rab5 endosomes, but this needs to be further tested. What happens to endocytic machinery in these conditions? Further analysis will strengthen the study.
- 4) Related to point 3, it would be ideal to have higher resolution imaging of the oenocytes in the experiments displayed in fig 5. These images are informative but quite small. Higher resolution insets and/or electron microscopy would better resolve the intracellular changes in these experiments.
- 5) The final section of the study examining Impl2 is interesting, and implies changes in dILP signaling at the FB and other organs in Desat1-RNAi. Can insulin signaling in the FB or elsewhere be more directly evaluated to further support these claims?

Reviewer #3

(Remarks to the Author)

The article is well written and nicely executed; therefore, I strongly suggest considering this manuscript for publication in Nature Communications after some essential revisions.

Major comments:

-I would start Figure 1 with a diagram explaining the fasting schedule the adult flies have been treated with. I would also add a panel B of an adult fly showing the location in the abdomen of the oenocytes studied in this article. In the text, I would also better explain whether there is continuity or whether the oenocytes and the fat body are physically separate. Thanks to Alex Gould's lab article, we know that the larval oenocytes are located in the epidermis. Further description in the text and a supporting figure/sketch showing the anatomical location in the adult would be helpful.

-It is clear that the new starvation protocol established in this work induces changes in lipid levels at the hemolymph level and at the whole larva level, which could be interpreted as changes mainly at the adipose tissue level. Although these changes are very relevant, the results shown in Figure 1 and extended Figure 1 deserve further explanation both at the text level and at the figure level. In addition, I've considered including the results shown in the extended figure 1b-c in the main results.

- I would combine the results in Figure 2e and f to show the saturation of several key lipids in OETs>Desat1RNAi hemolymph, particularly DAG and PE.

-Figures 4 and 5 clearly show that altered lipoprotein internalization (Fig. 4a-c) in OETs>Desat1 RNAi animals is associated with a rigid actin network and endosomal vesicles in oenocytes, and that actin suppression treatments by overexpression of Tsr (twinstar) or Limk1 (LIM domain kinase 1) knockdown (Fig. 5a) result in a strong reduction of Apolpp accumulation on the oenocyte surface (Fig. 5b), as well as a normalization of lipoprotein levels in the hemolymph (Fig. 5c,d). What is the effect of this "rescue" of Apolpp release on starvation resistance and total TAG levels? These experiments, which are fundamental, are shown at the end of the article in Figure 7 e and f. I think the figure should be rearranged to include these results in Figure 5, as this allows a much smoother flow of results. Moreover, it would be desirable that the images' quality and the markers (Hoestch, Apolpp, and phalloidin) used in Figure 4 be reused in the immunostains shown in Figure 5.

-This study demonstrates the relevant role of the oenocyte in the accumulation of lipids during periods of starvation, similar to that of the mammalian liver. Although this is cited in the Introduction, a paragraph highlighting the role of the oenocyte and its function as an analog of the mammalian liver, at least in fasting conditions, and interacting by signaling via Impl2 with the fat body, should be included in the discussion.

Reviewer #4

(Remarks to the Author)

In this manuscript authored by Li et al. we are presented with interesting insight into a rather complex and systemic processes that involve several organs to regulate lipid dynamic during nutrient restriction. The authors present very detailed and important description of lipidomic changes in the whole body of the fly compared to haemolymph. They demonstrate that similar as in mammals upon starvation there is a shift towards lipid desaturation and an increase in hydrocarbon length. They then investigate and confirm the participation of the oenocytes as a potential "liver-like" tissue, in addition to the fat body, in the processes that regulate the lipid shifts in haemolymph and affect fat body lipolysis. On molecular level they corroborate their findings by knocking down Desat1, fly desaturase, which results in the absence of starvation dependent lipid accumulation in oenocytes as well as in an increased accumulation of lipoproteins (likely derived from Fat body – not demonstrated in this paper) in intriguing rab5-positive structures wrapped in by actin at the cell cortex. The Desat1 deficient oenocytes seem to work as a trap for lipoproteins, here demonstrated on examples of apolpp and albumin, which they demonstrate can be released by inhibiting excessive actin polymerisation. The authors looked at the lipoproteins as a likely source for lipids that may be derived from the fat body following starvation, which they assume is true based on juxtaposed timely shifts in peak lipid amounts first in FB and then in OE. Furthermore, authors proposed that the OE likely also communicate with other tissues following starvation, by releasing lipids or proteins. They, identify Impl as one such factor and demonstrate that its release is prevented by desat1 deficiency and rescued by inhibited actin polymerisation.

Major point. In my view this paper offers great amounts of data and delivers important evidence that oenocytes indeed contribute to lipid shifts during starvation as a liver-like tissue. While doing so the authors, provide important insights into some key molecular players and pathways that may contribute to the crosstalk with the fat body and other tissues. However, in order to claim that these molecules are indeed are either derived from the fat body or will act on the fat body, additional work must be done, ie. using additional binary expression tools. To me it appears that the authors make many deductions when interpreting the experiments in the results section, particularly relating to OE and fat body cross talk, but not really demonstrating the validity of their statements by actually looking at the fat body or manipulating the fat body. The conclusions or interpretations should perhaps be revised with caution to avoid overstatements. Below, I am including some examples, excluding detailed revision of discussion:

P8 L6 The authors suggest that the apolipoproteins are FB derived. Can that be demonstrated – i.e. silencing apolpp in the fatbody and analysing effect on apolpp accumulation in OE>Desat1R. Alternatively, conclusion may be toned down in results section. All I see that the apolpp is not OE-derived.

P10 LL4-5 “suggesting that lack of ImpL2 secretion could lead to a starvation-like condition with stimulation of FB lipolysis.” – this is a valid speculation not demonstration. Therefore, statement further below in the same page LL33 does not seem valid to me: “5) ImpL2 controls survival resistance by controlling FB lipolysis”. Similarly, in L35: “promoting FB lipid storage”. P11 L 1 ie may read: “To question how exchange between BF and OE MAY BE carried out...”.

Minor points:

1. I would like to hear more of authors thoughts on possible relationship of observed phenotypes to autophagy flux in desat1 KD OE. To me it almost seems like the endolysosomal pathway may be stuck upon desat1 knockdown. How specific are then those effects that we see? Does the lack of lipids perhaps perturb the membrane biogenesis and then subsequently there is a block in autophagic flux? What if the apolipoproteins are digested via autophagy after uptake into OE? And if indeed autophagy is stalled maybe that's why we see them trapped at the surface?
2. To me it is not clear how brummer contributes directly to OE phenotypes. OE>desat1-IR; brummer-IR seem to have no rescue effect, brummer seems to be more important in toher tissue? Additionally, brummer rescue experiments using the classic allele were performed on a mutant TM3 balancer background, reflected in severely reduced half life of desat1 deficient and control flies in fig 3h compared to fig 2a,b. TM3 is a “brocken” chromosome containing multiple inversions, we don't know how these contribute to the phenotypes studied. In order to claim a “full rescue” by haploinsufficiency in brummer (p7 L13), control experiments without TM3 or demonstrating tm3 contribution to starvation sensitivity would be necessary. Same is true for the global TAG dynamics. This should be at least made more clear for educated interpretation of results by a non-Drosophilist.
3. Regarding gene selection. P9 L24 Could the authors perhaps specify on which basis the 8 genes were selected? Additionally, extended figure 9c – it remains unclear to me why ImpL was selected over the other genes – nplp2 seems to have similar dynamic during starvation.

Version 1:

Reviewer comments:

Reviewer #1

(Remarks to the Author)

I would like to thank the authors for all their efforts to clarify my questions on the previous manuscript.

The manuscript is really clear right now, and I think the authors very much improved their argumentation with additional experiments concerning the role of insulin signaling and also carbohydrate metabolism.

I have one minor point left: I would maybe indicate/label Desat1 in the graphical abstract (Figure 8) to make its role more obvious.

I am absolutely convinced that this manuscript is a very important work to dissect the role of oenocyte in energy metabolism regulation and especially their so far less established role in inter-organ crosstalk during energy mobilization. This work will be important for future studies elucidating energy mobilization and redistribution in stress responses such as infections and the trade off to immune response.

All my questions were answered and I would highly recommend the revised version for publication.

Reviewer #2

(Remarks to the Author)

The revisions has added a wealth of data and addresses the major concern. a XBP1-EGFP system more thoroughly examines ER stress. Numerous other concerns have been addressed either in the manuscript or in the very detailed point-by-point discussion letter. This is an important study and should be published.

Reviewer #3

(Remarks to the Author)

The appropriate changes have been made by the authors in response to my comments and the manuscript has been clearly improved. I have no further comments and therefore consider the manuscript suitable for publication in its current form.

Reviewer #4

(Remarks to the Author)

I thank the authors for addressing the raised questions in great detail. The additional data for Brummer and ImpL2 significantly enhance the clarity of the results presented in the manuscript, as well as their interpretation regarding bidirectional cross-talk between FB and OE in regulating lipidomic shifts during starvation. The research presented here certainly opens many more questions, i.e., on the relationship between Dsat-deficiency and endosomal pathway/actin rigidity and autophagy arrest. Thank you for confirming this experimentally. In my view, additional exploration of these questions is outside the scope of this study and will hopefully be addressed in more detail in the future. I strongly recommend this manuscript for publication.

Perhaps, some minor changes could improve the clarity of the presented results and reproducibility of the experiments

presented:

- P8L240-243 The statement regarding the accumulation of lipids in the gut of fed flies is unclear.
- Provision of sequence details for generated LexAop shRNA
- Perhaps authors may consider adding the unresolved questions related to disruption in endocytosis, autophagy, and actin rigidity upon OE>dsat-KD to the discussion, as a perspective on future research?

Point-to-point address for NCOMMS-24-79193 (responses in black)

We would like to thank all four reviewers and the editors for the overall positive evaluation of our manuscript. The comments and suggestions have been helpful and constructive. Based on this criticism, the revised version is much improved. Below are our responses to each point:

Reviewer #1:

In the manuscript „Desaturase-dependent secretory functions of hepatocyte-like cells control systemic lipid metabolism during starvation in *Drosophila*“, Li and colleagues explore the role of Desat1 in oenocytes for systemic regulation of lipid metabolism and lipolysis during starvation in larval and adult *Drosophila*.

The study provides new insights on the role of hepatocyte-like oenocytes in regulating energy distribution from lipid storage in the fat body during starvation. Here the authors give detailed mechanistic insights on the role of Desat1 in oenocytes as a key regulator of lipid metabolism by regulating soluble factors and their exocytosis, such as Impl2, but also by altering lipoprotein sequestration and endocytosis in oenocytes. Overall, the authors provide profound evidence that oenocytes are key regulators of energy mobilization and inter-organ communication with the fat body to modulate systemic lipid metabolism during starvation. The manuscript and findings are of high relevance to the field, whereas oenocytes are often neglected or underexplored in their role during stress responses and infection, where the organism has high needs in rapidly adjusting energy storage and mobilization.

The manuscript is well written and I really enjoyed reading it. While reading, a couple of questions came up which the authors could additionally test to extend their study and make their findings even more solid before publication.

Thank you for the kind words and interest in our work.

1) In the beginning, when the authors describe their new NR model, it is unclear which fly line is used, please clarify. Furthermore, I think it is important to show the overall survival of the flies on NR in Figure 1a and Ext. Figure 1a.

We completely agree that providing detailed genotype information and overall survival data is essential for establishing the experimental foundation of our study. In response to these suggestions, we have made the following revisions:

We have now provided the genotype information for the strain used to describe our new NR model in the legend of Fig. 1b. Moreover, we have conducted overall survival experiments to establish baseline physiological parameters for our experimental flies. The survival curves demonstrate a longer life time of flies with our modified starvation protocol. These new data have been incorporated as the Figure 1b and Extended Data Fig. 1a, replacing the previous versions.

2) Temperature plays critically into the regulation of metabolic activity in flies. If possible, I recommend to

show NR wildtype survivals and the induced changes in lipid metabolism at 25°C and in the same direction, it would be good to show the effects of *Desat* KD maybe with an auxin inducible Gal4 approach in parallel.

We sincerely thank the reviewer for bringing up the role of temperature on the metabolism of flies. Therefore, we carefully evaluated multiple inducible approaches at the onset of this study, including both temperature-sensitive and drug-inducible methods. Our decision to employ the GAL80ts system was based on the following scientific and technical considerations:

1. The temperature-inducible GAL80ts system provides temporally precise and spatially uniform knockdown across all experimental flies. As drug-inducible systems require compound consumption through food intake, we worried that the individual flies may consume different amounts of the inducing agent, leading to heterogeneous knockdown efficiencies both within and between experimental groups.
2. To our knowledge, auxin-inducible GAL4 systems with oenocyte-specific expression were not available
3. We noticed that PromE-GeneSwitch-GAL4 flies are not viable in the homozygous state. For the revision, we nevertheless performed starvation assays using PromE-GeneSwitch-GAL4 flies to directly compare with our GAL80ts results obtained at 29°C. At 25°C, we observed a statistically significant and profound reduction in starvation survival in the RU486-treated *Desat1* knockdown flies compared to the untreated flies (new Extended Data Fig. 2a). This finding demonstrates consistency with our primary results obtained at 29°C using the GAL80ts system.

3) One of the most interesting findings is that the data provided here highlights again the essential temporal regulation of lipid metabolism in the fat body. As the authors pointed out, before lipolysis is initiated in the fat body and the oenocytes start sequestration of lipoproteins (but also in parallel), systemic carbohydrate metabolism and insulin signaling is extremely adjusted and important for efficient energy mobilization. On nutrient restriction it would be helpful if the authors could provide additional insights on the regulation of carbohydrates and insulin signaling (activity on protein level (pAKT and foxo TF translocation and /or regulation of foxo targets) at different stages of NR and *Desat* KD in oenocytes.

Thank you for this insightful question regarding the broader metabolic context of our findings. The overall logic of our study is that the FB increases lipolysis in starvation. Hence, *bmm* is upregulated in starvation. However, a too rapid depletion of energy stores can also be detrimental in prolonged starvation. This is why removal of one *bmm* copy rescued the starvation sensitivity of $OE^{ts} > Desat1^{RNAi}$ animals. We could further show that oenocytes seem to regulate FB lipolysis by secreting factors such as *ImpL2*. But it remained unclear whether this involves insulin signaling. Also, the effect of carbohydrates was not yet studied.

We therefore addressed the role of carbohydrates by performing glycogen measurements. We observed a gradual reduction in glycogen levels during starvation in the control group, with approximately 75% depletion by day 3 of starvation. On both day 1 and day 3 of starvation, glycogen levels were significantly reduced in both *ImpL2* and *Desat1* knockdown groups compared to controls. Interestingly, the *ImpL2* overexpression group did not show obvious glycogen reduction during the early stages of starvation. Since we found an increased level of glycogen in the *ImpL2* overexpression group at day1 and day3, we also

checked glycogen level at later stages of starvation. Interestingly, the glycogen level of ImpL2 overexpression fly at day 5 of starvation is similar to the control flies at day 3 of starvation. These new results can be found in Fig.7e,f.

Regarding insulin signaling, we first performed starvation sensitivity experiments upon insulin receptor (InR) manipulation in the fat body or oenocytes. These showed that when silencing the *insulin receptor (InR)* in FB and oenocytes, an enhanced survival was observed in our starvation protocol, while the opposite was the case for *InR* overexpression (these new results can now be found in Fig.7K and Extended Data Fig. 13c). This correlated with increased and decreased TAG levels, respectively (new Fig.7k and l). Interestingly, we did not see this kind of correlation when *InR* was manipulated in the FB (New Extended Data Fig. 13d). Together, this suggests that IIS suppression, particularly in the oenocytes, is beneficial for prolonged starvation.

As requested, we also analyzed foxo translocation using immunostaining in fed and NR conditions, but we could not find much difference in foxo localization. Moreover, we performed qPCR for foxo targets using whole animal and cuticle samples. In the whole animal, we found that *bmm* is elevated in NR and even more in *Desat1* KD (new Fig.3f, g), while *4EBP* increases in *Desat1* KD but not in NR conditions. In the cuticle which contains different tissues, including FB, oenocytes, muscle and epidermis, *bmm* went up and *4EBP* went down in starvation (R_Fig. 1), suggesting that these two foxo targets are not regulated in the same direction. When reanalyzing our snRNA-seq results, we similarly noticed that during starvation the IIS targets *bmm* and *4EBP* are regulated in the opposite manner in FB cells: while *bmm* was upregulated during starvation in FB cells, *4EBP*, another foxo target, was downregulated. There were also cell type-specific differences for *4EBP* during starvation, as it was reduced in the fat body and increased in oenocytes. *InR*, another foxo target, showed only very minor changes in FB and oenocytes when comparing fed vs. starved samples. This shows that whole animal or even cuticle samples which both are a mix of many tissues do not adequately reflect foxo activity. A possible explanation for this is that dILPs are present in both fed and starved conditions and that they have tissue-specific effects. Especially dILP6 might act locally between FB and oenocytes to maintain IIS in these cells. As requested, we also performed pAKT Western blotting on whole animal lysates. But as we had problems with the non-phospho Akt Western blot that we needed for the normalization the results were not conclusive. Due to the abovementioned limitations, we decided to neither include the whole animal and cuticle foxo target qPCR nor the pAKT WB in the revised version. The foxo IF was included to demonstrate that ImpL2 can indeed act as an antagonist of IIS. Furthermore, we included foxo target expression from the snRNA-seq data to demonstrate the cell type-specific effects. These results can now be found in Figure 7, Extended Data Fig. 12 and 13 and are described on page 11.

R_Fig.1 4EBP(Thor) mRNA level normalized by RPL32 in whole fly sample or cuticle at fed or starvation condition, n=3.

4) *Impl2* is a negative regulator of insulin signaling therefore it would be interesting to understand the regulation of insulin signaling by *Impl2* is seen in the fat body and affects

We thank the reviewer for this question regarding the role of *Impl2* in *Drosophila* insulin/insulin-growth factor signaling (IIS). The functional characterization of *Impl2* in insulin signaling regulation has indeed been the subject of considerable scientific debate. The traditional view, supported by extensive literature including Honegger et al. (PMID: 1841298), positions *Impl2* as a functional homolog of mammalian insulin-like growth factor binding proteins (IGFBPs) that acts as a systemic antagonist of insulin/IGF signaling by sequestering circulating dILPs. However, recent studies have challenged this simple antagonistic framework. Ghosh et al. (PMID: 35316653) presented compelling evidence suggesting that *Impl2* may facilitate dILP uptake by the corpora cardiaca, potentially serving as a carrier protein rather than merely an inhibitor. Recent structural studies have shown that *Impl2* can include a R-state conformation of insulin and IGFs which is considered to be closer to the active form of these hormones (PMID: 30242155). The interpretation of *Impl2* function is further complicated by the inherent complexity of the *Drosophila* insulin system, which differs substantially from mammalian models. *Drosophila* expresses multiple insulin-like peptides with distinct tissue-specific expression patterns, temporal regulation, and potentially different antagonists or agonists. Finally, also IIS-independent functions of *Impl2* in the regulation of energy stores have been described (PMID: 33046910). This complexity necessitates careful experimental design and direct cellular readouts to accurately assess insulin pathway activity. It should also be emphasized that we are using a novel protocol for prolonged starvation, making direct comparisons with other studies difficult.

To directly evaluate insulin pathway activity under *Impl2* manipulation, we employed immunofluorescence analysis of foxo subcellular localization. Our immunofluorescence analysis revealed a significant reduction in nuclear foxo signal intensity in oenocytes from *Impl2* knockdown flies compared to controls and *Impl2* overexpression groups. This reduced nuclear foxo localization indicates enhanced insulin signaling activity in the absence of oenocyte *Impl2*, which is consistent with the traditional model whereby *Impl2* functions as an insulin signaling antagonist. As a positive control, we used *InR* KD in oenocytes. These new results can be found in Extended Data Fig. 13a.

We further show that the starvation phenotypes for oenocyte-specific *ImpL2* KD and overexpression strongly resemble those of oenocyte-specific *InR* overexpression and knockdown, respectively. Also, this supports the antagonistic role of *ImpL2* in the IIS. Interestingly, the correlation with the respective changes in TAG and glycogen levels (high in *InR* knockdown and low with *InR* overexpression) were much higher when *InR* was manipulated in the oenocytes than in the FB (new Figure 7 and Extended Data Fig. 13). Whether this also reflects autocrine activity of *ImpL2* in the oenocytes will need to be addressed in more detail in future studies.

5) Is the decreased release of *Ilp2*, *3*, *5* from IPCs in the brain affected by *Desat1* KD or overexpression of *ImpL2* during NR (more a control experiment to prove that this is not the case)?

We checked the mRNA level of *Ilp2*, *3*, *5* in the adult heads upon *ImpL2* KD and overexpression as well as between fed and starvation conditions. Interestingly, we found an increase in the mRNA of all three *Ilps* in the *ImpL2* overexpression group (R_Fig. 2). This is in line with a previous publication (PMID: 21108726) reporting on a negative feedback loop upon *ImpL2* overexpression.

R_Fig.2 *Ilp2*, *3*, *5* mRNA level normalized by RPL32 in adult heads sample at fed or starvation conditions, n=3. Statistical tests: two-way ANOVA.

Second, we performed Western blotting to check dILP2 proteins levels in hemolymph (for the other two Dilps no antibody was available). Interestingly, we found dILP2 protein levels in the *ImpL2* overexpression group that match the *Ilp2* mRNA levels in the head (R_Fig.3). Also, dILP2 protein in the *ImpL2* KD correlated with the mRNA, confirming the negative feedback loop also on the protein level. As in the abovementioned publication, that also shows an increase in lifespan and TAG stores upon *ImpL2* overexpression (PMID: 21108726), it is unclear whether this feedback loop influences the role of oenocyte-derived *ImpL2* during starvation. Given that the *ImpL2* KD does not show nuclear foxo, we suspect that it might not be the case. However, due to these uncertainties we decided not to include these results in the revised version.

R_Fig.3. Western blot analysis of Ilp2 in the hemolymph between fed and starvation conditions. n=3

6) The authors suggest lipid flux between oenocytes and fat body during NR. I would suggest here to do ^{13}C -FFA feeding before NR onset to monitor the precise flux of lipids and also to better determine the dynamics seen in the lipidomics and to provide the final proof that lipids are fluxing from the fat body to oenocytes and back in distinct lipid species.

We thank the reviewer for this excellent suggestion. The proposed ^{13}C -fatty acid labeling approach is indeed powerful and could provide direct biochemical evidence for inter-tissue lipid trafficking, particularly the hypothesized exchange between the fat body and oenocytes. However, due to the small size and embedded anatomical position of oenocytes, achieving the required tissue purity for accurate mass spectrometry analysis is highly challenging. Moreover, as discussed in the next point, fat body isolation under starvation conditions is technically problematic. Altogether, we feel that the establishment of the ^{13}C -incorporation in the limited amounts of material obtainable from these two tissue types is beyond the scope of this manuscript, especially since this method depends on dietary uptake of ^{13}C -fatty acids, while our interest is the prolonged starvation.

7) The time-resolved lipidomics of hemolymph and whole fly are very interesting. However, I think it would be interesting to answer how the lipid composition in the fat body (and oenocytes) are changing. Whereas oenocytes might be tricky in lipidomics, but analysis of dorsal abdominal fat body could give here more insights.

We thank the reviewer for this valuable suggestion concerning tissue-specific lipidomic analysis of the dorsal abdominal fat body and oenocytes under both fed and starved conditions. We fully acknowledge that such comparative profiling could provide important mechanistic insights into tissue-specific metabolic specialization and potential inter-tissue lipid trafficking during nutritional stress. Hence, we carefully evaluated the feasibility of this approach. However, similar to the experiment raised in point 6, we encountered substantial methodological obstacles.

These challenges primarily arise from the close anatomical association between the fat body and oenocytes, which is further exacerbated under starvation. Even under optimal fed conditions, achieving complete separation of the dorsal abdominal fat body from oenocytes remains technically demanding, as oenocyte clusters are frequently co-isolated with fat body tissue. Through careful optimization of microdissection techniques, we were able to obtain relatively pure fat body samples in the fed state. In

contrast, starvation causes a pronounced reduction in fat body cell volume and overall tissue mass. Additionally, while in the fed state these tissues remain relatively mobile within the hemolymph cavity, starvation renders them firmly attached to the body wall. This not only makes physical separation extremely difficult without causing substantial damage, but also compromises tissue integrity, leading to fragmentation during dissection and introducing potential artifacts into downstream lipidomic analyses.

8) At some point in the manuscript, the authors state that the *Desat1* in oenocytes is important for lipid storage in the neighbouring fat body. Here, the authors only provide evidence for this in late stages of NR, so I would be specific that this is during NR and not steady state or test if *Desat1* in steady state might also affect TAG storage and LD formation in fat body. Furthermore, I am not convinced yet that this is only specific to neighboring fat body, the authors should test in adult *Drosophila* if *Desat1* in oenocytes will also affect distant fat body for example in the head behind the optical lobes.

Sorry for the confusion, we will specify our statement that *Desat1* in oenocytes is important for lipid storage during NR (Page 7, line 20). Moreover, we thank the reviewer for the suggestion regarding organism-wide fat body analysis. We have now analyzed the fat bodies in the head with oenocyte-specific *Desat1* manipulation, allowing us to distinguish between local paracrine effects and systemic metabolic coordination.

We initially attempted direct dissection of cephalic fat body tissue behind the optical lobes followed by BODIPY staining. However, the small size and delicate nature of the cephalic fat body resulted in frequent tissue loss during dissection. We subsequently tried cryosectioning of whole adult heads, but maintaining consistent anatomical positioning across samples proved technically challenging for reliable quantification. In the figure below, two different sections from the same head are shown to demonstrate this problem (R_Fig.4).

R_Fig.4. Different layers of section exhibit a distinct pattern of BODIPY staining. Green: BODIPY, Blue: Hoechst.

Consequently, we performed biochemical quantification of total TAG levels from adult heads across different experimental groups and time points. Using this approach, we could show that head samples in the control group exhibited a reduction of TAG level during starvation. Both oenocyte *Desat1* KD groups showed significant reduction in cephalic TAG levels during starvation compared to the control, which directly parallels our whole-body TAG measurements (new Extended Data Fig. 3i). These results demonstrate systemic metabolic effects extending beyond the abdominal fat body to distant fat depots.

9) Can the authors elaborate a bit further on the gut during NR and lipid dynamics here. Are there any effects during NR compared to “standard” starvation? Does *Desat1* KD or *Impl2* OE also affect the gut, especially in terms of lipoprotein sequestration and lipolysis in this organ at early stages of starvation.

We sincerely thank the reviewer for this excellent suggestion. The gut is indeed a critical metabolic interface for lipid absorption, processing, and lipoprotein trafficking, making it highly relevant for understanding the systemic metabolic network coordinated by oenocytes.

Our lipid analysis in the fed state revealed pronounced lipid accumulation in the R3 and R5 intestinal regions of oenocyte *Desat1* knockdown flies (new results provided in Extended Data Fig. 5a). Together with the hemolymph Western blots, this observation offers important independent validation of our proposed lipoprotein sequestration mechanism. Notably, the phenotype we observed closely resembles that reported by Palm et al. (PMID: 22844248; PMID: 39235946) in fat body-specific *apolpp* RNAi flies. Given that *apolpp* encodes an essential component of lipoprotein particles, the similarity between our gut phenotype and the established *apolpp* knockdown results provides strong support for our conclusion that oenocyte *Desat1* dysfunction can cause systemic lipoprotein sequestration.

10) The authors state in their manuscript: “Together with the strong LD decline in the FB of OETs> *Desat1*RNAi animals, this indicates that oenocytes non-autonomously promote lipid storage in the FB, presumably by suppressing lipolytic activity”. From my point of view this has to be either stated specifically for starvation/NR or it has to be shown that this is also the case during steady state (I am not sure if I can see or interpret that from the data provided in the fed stage with *Desat1* KD)

We completely agree that oenocyte *Desat1* knockdown flies exhibit significantly more pronounced lipid modification and turnover phenotypes in starved vs. fed conditions. This is particularly obvious in Fig.3 d,e. In the revised version, we have highlighted this difference between starved vs. fed conditions more clearly (page 7, line 19).

11) Is *Impl2* the only adipokine or factor that is affected by changes in exocytosis upon *Desat1* KD in oenocytes?

Our results suggest that the secretion defect in *Desat1*-deficient oenocytes is not limited to a specific protein but rather reflects a general impairment. We hypothesize that the actin and endosome accumulation at the cortex creates a physical barrier, preventing *Impl2*-containing exocytic vesicles from fusion with the plasma membrane. Consistently, secretion of transgenic human albumin was also disrupted

(Fig. 6a). This implies that the secretion of other factors upregulated in oenocytes during starvation may likewise be compromised.

To further explore this possibility, we performed starvation assays and TAG measurements following knockdown of additional secreted proteins (Extended Data Fig. 12c), which revealed varying degrees of effect. While our current study focuses on *ImpL2*, we plan to systematically investigate other oenocyte-secreted factors under diverse metabolic challenges (e.g., high-fat and high-sugar diets) in order to generate a comprehensive map of the oenocyte secretome and its physiological functions.

12) Is ILP6 signaling from the fat body to oenocytes affected by *Desat1* KD in oenocytes?

We sincerely thank the reviewer for their question regarding the role of dILP6 in lipid metabolism and the interplay between the fat body and oenocytes. To investigate the role of dILP6, we performed quantitative PCR (qPCR) to measure the mRNA levels of *Ilp6* in cuticle samples, which contain several tissues including the fat body, oenocytes, and muscle (R_Fig.5). Our results indicate an unchanged expression of *Ilp6* in both oenocytes *Desat1* and *ImpL2* knockdown groups. However, we acknowledge two key limitations in interpreting these results:

The mixed tissue composition of cuticle samples may introduce variability in *Ilp6* levels, particularly due to changes in fat body mass (even though we used cuticles containing mainly fat body and oenocytes). Moreover, *Ilp6* mRNA may be subject to positive or negative feedback loops, as we can see with other *Ilps* (see comment #5). Unfortunately, we could not measure *Ilp6* protein as there is no specific antibody for *Ilp6* available.

R_Fig.5 *Ilp6* mRNA level normalized by RPL32 from adult heads sample at starvation condition, n=3. Statistical tests: one-way ANOVA, no significant changes between groups.

We also found an increased level of *Ilp6* in flies with oenocyte-specific *ImpL2* overexpression (R_Fig.6a). Because Chatterjee et al. (PMID: 25472843), reported the importance of *Ilp6* secreted from the fat body. We performed a survival experiment using flies with fat body *Ilp6* knockdown or overexpression. Interestingly, we found that both groups exhibited a reduction of lifetime (R_Fig.6b), which is also

consistent with Chatterjee et al. Next, we checked the mRNA level of *Ilp6* in the *dilp6* overexpression group by using qPCR and found a strong up-regulation of *Ilp6* in the cuticle samples (R_Fig.6c).

R_Fig.6 a *Ilp6* mRNA level normalized by RPL32 from adult cuticle samples at starvation condition, n=3. Statistical tests: Unpaired t test, *: p<0.05. **b** Starvation sensitivity of flies with *Ilp6* KD or overexpression in the fat body specifically. n=50, p value was calculated using the Log-rank (Mantel-Cox) test. ****: p<0.0001, ns: non-significant. **c** *Ilp6* mRNA level normalized by RPL32 from adult cuticle samples at starvation condition, n=3. Statistical tests: Unpaired t test, **: p<0.01.

These results indicate that dILP6 signaling from the fat body to oenocytes is crucial for starvation sensitivity, and maintaining an appropriate balance of dILP6 levels is essential for starvation tolerance. However, as we believe that the FB-derived dILP6 is not the focus of the paper we decided to not include the results in the revised version.

13) The effects of *Desat1* KD in fat body are interesting and showing profound changes during development, however somehow do not support the storyline of the manuscript. The authors should consider inducible *Desat1* KD in fat body during NR in adult flies similar to the experiments with oenocyte-specific KD.

Thank you for this comment. The reason to study *Desat1* in the FB was to validate the LD phenotype we observed in the oenocyte in a cell type that is specialized in LD formation. As requested, we have now also studied *Desat1* function in adult FB using an inducible system. We performed several experiments with the GAL80ts-inducible fat body driver: 1. Immunofluorescence analysis with BODIPY (lipid droplets) and phalloidin (fat body architecture); 2. Survival assays under starvation conditions; 3. TAG content measurement. Together, the results clearly support an important role of *Desat1* in TAG and LD formation. They further provide an oenocyte-independent validation of the importance of maintaining sufficient TAG stores in our prolonged starvation assays. These new findings have been added to Extended Data Fig. 3g-i and Fig. 13d.

14) Minor point: Figures should appear in numeric order in the text. Figure 1g is missing the labeling where the higher magnification is taken from.

Thank you. We corrected Fig. 1g (now Fig. 1k).

15) Minor point: In Figure 6g it looks like there are only two samples in a group at day 1, but statistics were applied, please remove the stats or add samples to the analysis.

Thank you. We added one additional sample and corrected the statistics.

16) For all bar graphs: please indicate individual samples with dots on top of the bars, as done in Figure 5 and 6.

Thank you. This was done.

17) Please provide a scale bar for the higher magnification inserts in some of the imaging figure panels.

Thank you. This was done.

Reviewer #2:

This study investigates the roles of *Drosophila* liver-like oenocytes in lipid metabolism and interorgan lipid transport. It begins with a lifespan analysis and lipid analysis of flies fed an ion-enriched starvation diet, showing a lifespan extension and changes in lipid storage, particularly in females. Lipid changes include an increase in double bond lipids, suggesting an increase in lipid desaturation. They find that depletion of the *Drosophila* fatty acid desaturase *Desat1* specifically in oenocytes leads to changes in the lipid profile of the blood and reduced fat storage in the adipose-like fat body. *Desat1* depletion also causes more starvation sensitivity. There were also changes in oenocyte cortical actin and in lipoprotein secretion into the blood. Insulin influencing protein ImpL2 (mammalian IGFBP7) secretion from oenocytes is also reduced with *Desat1* knockdown.

This work is important and nicely dissects how oenocytes significantly impact the lipid profile of secreted lipids in the animal hemolymph. It also investigates the relationship between lipid starvation profile and aging, as well as cold tolerance and other stresses like nutrient restriction. The analysis on lipid transport and the actin cytoskeleton is also intriguing, although additional work is needed to provide mechanistic insights. In general, the study is well executed, although some conclusions need additional data to be supported by the observations. A general concern is that the changes observed with *Desat1* depletion in oenocytes could result in general ER stress or other cellular stresses, and thereby phenotypes may represent pleiotropic effects.

Thank you for finding our results important and intriguing.

1) A general concern is the *Desat1* RNAi depletion in the oenocytes can lead to many changes in these cells in addition to lipid saturation changes. More saturated lipids can lead to ER stress and perturbed

organelle homeostasis for many organelles including mitochondria. Determining whether ER stress is elevated in oenocytes in *Desat1* RNAi, and whether mitochondrial function is perturbed, are important controls that provide context for this investigation.

We sincerely thank the reviewer for these excellent suggestions regarding lipotoxicity and organelle dysfunction in *Desat1*-depleted oenocytes. These insights are particularly valuable given that higher lipid saturation can indeed cause cellular stress. We already show a number of cell-autonomous effects of *Desat1* KD in the previous version. The most extreme effect was the nearly complete depletion of adipose tissue when expressing *Desat1* RNAi in the larval fat body. For the oenocytes, we have now performed a number of additional analyses of organellar stress and morphology.

For ER stress analysis, we utilized the *XBP1-EGFP* reporter system (PMID:17170705) to directly assess ER stress in *Desat1* knockdown oenocytes. Our findings reveal stage-specific ER stress responses: when expressing *Desat1* knockdown in larval oenocytes by shifting to 29°C in the larval stages, we found a significant increase in nuclear XBP1-EGFP signal, consistent with earlier results from our lab using mammalian cells (PMID: 35550039). However, when performing our usual expression protocol for adult oenocytes, the XBP1-EGFP signal was predominantly cytoplasmic rather than nuclear (R_Fig.7). As we are not sure how to explain it, we did not include these results in the revised version. We also stained for Phospho-eIF2 α which is a marker for cellular stress, including ER stress, but could not find significant differences, making the interpretation of UPR responses even more complicated (R_Fig.8).

R_Fig.7 Representative immunofluorescence images of XBP1-GFP in larva or adult. In larva oenocytes with *Desat1* depletion, we found an increased XBP1-GFP signal in the nucleus. In adult oenocytes, we observed an increased XBP1-GFP signal but not in the nucleus, n=3. Scale bars: 200µm in larva and 20µm for adult.

R_Fig.8 Representative immunofluorescence images of Phospho-eIF2 α in oenocytes with or without Desat1 KD, n=2, scale bars: 20 μ m.

In addition to our ER stress marker analysis, we examined the morphology of the endoplasmic reticulum and different Golgi compartments in Desat1-depleted oenocytes using confocal microscopy. Here, we could not find significant differences (new Extended Data Fig. 10).

For the mitochondrial analysis (R_Fig.9), we assessed mitochondrial morphology and function. Our results suggest mild changes of mitochondrial morphology using the mito-GFP marker (BDSC_8442). Mitochondria appeared less elongated. However, when using Mitotimer (BDSC_57323) to assess mitochondrial turnover and Mito-QC (BDSC_91641) to assess mitophagy, we could not detect significant differences.

R_Fig.9 Representative immunofluorescence images of mito-GFP, Mitotimer or Mito-QC in oenocytes with or without *Desat1* KD, n=3, scale bars: 20um.

Together, these results show that there are a number of cell-autonomous effects induced by *Desat1* KD while some organelles appear surprisingly intact. The “strongest” effect is in our opinion the cortex alteration with the accumulation of actin and lipoprotein-containing endosomal vesicles, which may be responsible for the secretion defect seen in the *Desat1*-deficient oenocytes. However, we added a sentence to acknowledge potential other reasons for the secretion defect (p. 9, line 34-35).

2) *Desat1* RNAi in oenocytes resulted in reduced LDs in the FB over time. This is proposed as due to an increase in lipolysis in the FB (Fig 3e), but lipid biosynthesis may also be altered. Can lipolysis in the FB be investigated further? Does *Bmm* loss in the FB in this condition rescue the LD levels? A heterozygous *bmm*(1) mutant is utilized in Fig3g,h, but additional analysis is required to make this conclusion.

We sincerely appreciate the reviewer's insightful question regarding fat body lipolysis, which is a key aspect in our study.

In our manuscript, we demonstrated accelerated lipolysis in oenocyte-specific *Desat1* knockdown flies, evidenced by decreased total TAG levels and BODIPY staining across various starvation time points (Fig. 3a-

d). Additionally, removing one copy of *bmm* rescues starvation sensitivity and leads to elevated TAG levels during starvation, reinforcing the link between lipolysis and metabolic regulation (Fig. 3i, j). By contrast, we observed no rescue of TAG level when we silence both *bmm* and *Desat1* in oenocytes (Fig. 3h). Therefore, we conclude that oenocyte lipolysis is not the reason for the drop of TAG in the *Desat1* KD flies.

To further explore this, we initially planned to dissect adult fat bodies under starvation conditions and perform qPCR to assess *bmm* mRNA levels. However, due to the reduced size of the fat body and its tight adhesion to the cuticle, obtaining sufficient purified tissue for analysis proved challenging. As an alternative, we dissected the cuticle, which includes fat body, oenocytes, muscle, and other tissues, allowing us to evaluate *bmm* expression in a broader metabolic context. In cuticle samples, we found an increased *bmm* expression with the time of starvation and a significantly increased *bmm* level in $OE^{ts}>Desat1^{RNAi}$ flies. When using whole animal samples, we also saw an upregulation of *bmm* in these flies. These new results which explain the drop in TAG levels in the $OE^{ts}>Desat1^{RNAi}$ flies were included into Fig. 3.

In addition, we employed the LexA-LexAop system in conjunction with the UAS-GAL4 system to manipulate *bmm* expression specifically in the fat body while simultaneously targeting *Desat1* in oenocytes within the same animal. This dual genetic manipulation confirmed the initial rescue experiment with *bmm* allele but this time in a FB-specific manner (new results can be found in Fig. 3k,l).

We believe that these combined analyses have provided more insight into the interplay between fat body and oenocyte functions during starvation.

3) *Desat1* RNAi in oenocytes led to increased Apolpp lipoprotein accumulation at the oenocyte cell surface together with cortical actin. This is a key new observation and suggests actin dynamics influence lipoprotein lipid delivery at oenocytes. How does directly modulating oenocyte actin pools by depletion of other actin components of associated proteins impact lipid delivery to these cells? The implication is that actin promotes endocytic uptake into Rab5 endosomes, but this needs to be further tested. What happens to endocytic machinery in these conditions? Further analysis will strengthen the study.

Thank you for this comment. To test if actin promotes the endocytic uptake of lipoprotein we performed several genetic manipulations.

First, we analyzed, as requested, whether the depletion of actin components impact lipid delivery to oenocytes. We used the same RNAi line for LIMK1 that we employed for the rescue of the *Desat1* KD phenotypes to study neutral lipid content of oenocytes. However, when used alone we could not detect a phenotype in fed and starved conditions (R_Fig.10). When overexpressing an active form of *tsr* in the oenocytes, we could see a rather dramatic LDs accumulation in the fed condition, which may suggest increased lipid uptake (R_Fig.11).

R_Fig.10 Representative immunofluorescence images of LDs in oenocytes with or without *LIMK1* KD during fed or starvation conditions, n=2, scale bars: 20um.

R_Fig.11 Representative immunofluorescence images of LDs in oenocytes with or without *Tsr(active)* over-expression during starvation, n=3, scale bars: 20um.

Next, in an effort to increase actin levels in oenocytes, we silenced three distinct negative regulators of actin polymerization—*tsr*, *capping protein alpha (cpa; CAPZA* ortholog), and *slingshot (SSH* ortholog)—and also overexpressed LIMK1. In all these manipulations, we were unable to induce a significant increase in actin. For *tsr* knockdown, we verified that the RNAi line was effective by demonstrating robust actin accumulation in another cell type (nephrocytes). Why actin accumulation cannot be induced in oenocytes remains unclear. Overall, these negative results prevented us from drawing definitive conclusions regarding whether actin accumulation itself is sufficient to drive lipoprotein sequestration in oenocytes.

R_Fig.12 Representative immunofluorescence images of LDs in oenocytes with slingshot RNAi, Tsr RNAi, cap RNAi or LIMK1 overexpression, n=2, scale bars: 20um.

As requested, we also performed a more thorough investigation of the endocytosis pathway. Previously, we had noted an accumulation of Rab5 associated with lipoproteins, and phenotypes resembling Lpp and actin accumulation were observed upon *Rab5* knockdown, aligning with earlier studies (PMID: 17609110).

To further analyze the endocytic pathway, we performed knockdowns of the recycling endosomal Rab GTPase, *Rab11*, and the late endosomal/lysosomal Rab GTPase, *Rab7*. Our results showed that while *Rab11* knockdown led to a similar apolpp accumulation as the *Rab5* and *Desat1* KDs, *Rab7* knockdown was unremarkable (similar to knockdown of the autophagosomal regulator *Atg1* (see reviewer 4, point #1)). These findings suggest that it is mainly the early endosomal and recycling pathway that is affected in *Desat1* knockdown oenocytes. These new results can be found in Extended Data Fig. 7 and 8.

4) Related to point 3, it would be ideal to have higher resolution imaging of the oenocytes in the experiments displayed in fig 5. These images are informative but quite small. Higher resolution insets and/or electron microscopy would better resolve the intracellular changes in these experiments.

Thank you for this suggestion. We have now rearranged the images in this figure to enhance clarity and improve readability for the readership.

5) The final section of the study examining *ImpL2* is interesting, and implies changes in dILP signaling at the FB and other organs in *Desat1*-RNAi. Can insulin signaling in the FB or elsewhere be more directly evaluated to further support these claims?

To elucidate the role of insulin signaling in oenocytes following *ImpL2* manipulation, we conducted a comprehensive series of experiments. Please also refer to our responses to comments #4 and #5 by reviewer 1.

First, to determine whether insulin signaling activity changes upon *ImpL2* manipulation, we performed foxo immunostaining and analyzed its subcellular localization. We observed reduced nuclear foxo localization in the *ImpL2* knockdown group (Extended Data Fig. 12a), indicating elevated insulin signaling activity, which is consistent with previous findings (PMID: 39235946).

Second, we examined the mRNA levels of *Ilp2*, *Ilp3*, and *Ilp5* in adult heads under conditions of *ImpL2* knockdown or overexpression, as well as in fed versus starved states. Interestingly, *ImpL2* overexpression led to an increase in the mRNA levels of all three *Ilps* (R_Fig. 2). This observation is consistent with a previous report (PMID: 21108726) describing a negative feedback loop triggered by *ImpL2* overexpression. Next, we performed Western blotting to assess dILP2 protein levels in the hemolymph (antibodies for the other two dILPs were not available). Consistent with the mRNA data from adult heads, dILP2 protein levels were elevated in the *ImpL2* overexpression group and reduced upon *ImpL2* knockdown (R_Fig. 3). These results confirm that the negative feedback loop extends to the protein level. Currently, it is unclear whether this feedback loop influences the role of oenocyte-derived *ImpL2* during starvation. Given that the *ImpL2* KD does not show nuclear foxo, we suspect that it might not be the case.

Third, we manipulated the insulin receptor (*InR*) in both the fat body and oenocytes. Knockdown of *InR* in either tissue significantly extended lifespan under starvation, whereas *InR* overexpression in oenocytes markedly reduced survival. By contrast, *InR* overexpression in the fat body caused only a modest decrease in lifespan. *InR* knockdown in oenocytes also showed higher TAG levels in starvation, similar to *ImpL2* overexpression, while none of the *InR* manipulations in the fat body (*InR* RNAi, *InR* and *InR* DN overexpression) correlated with TAG levels. Together, these results indicate that reduced insulin signaling

enhances starvation resistance, with particularly strong effects mediated by oenocytes (New results can be found in Fig. 7k and Extended Data Fig. 13c,d).

In summary, our data collectively demonstrate that reduced insulin signaling, achieved through either *InR* knockdown or *ImpL2* overexpression, enhances starvation resistance, particularly through effects in oenocytes. These findings contribute to our understanding of tissue-specific insulin signaling modulation and its impact on metabolic adaptation to nutrient stress.

Reviewer #3:

The article is well written and nicely executed; therefore, I strongly suggest considering this manuscript for publication in Nature Communications after some essential revisions.

Thank you for these kind words.

Major comments:

-I would start Figure 1 with a diagram explaining the fasting schedule the adult flies have been treated with. I would also add a panel B of an adult fly showing the location in the abdomen of the oenocytes studied in this article. In the text, I would also better explain whether there is continuity or whether the oenocytes and the fat body are physically separate. Thanks to Alex Gould's lab article, we know that the larval oenocytes are located in the epidermis. Further description in the text and a supporting figure/sketch showing the anatomical location in the adult would be helpful.

Thank you for this suggestion. We have now added the requested schematic diagrams in Figure 1 with the fasting schedule, the inducible expression as well as the location of adult oenocytes.

-It is clear that the new starvation protocol established in this work induces changes in lipid levels at the hemolymph level and at the whole larva level, which could be interpreted as changes mainly at the adipose tissue level. Although these changes are very relevant, the results shown in Figure 1 and extended Figure 1 deserve further explanation both at the text level and at the figure level. In addition, I've considered including the results shown in the extended figure 1b-c in the main results.

We have now made it clearer that these are whole animal samples (p.5, line 19). In addition, we have moved the extended figure panels 1b-c into Figure 1 of the main results.

- I would combine the results in Figure 2e and f to show the saturation of several key lipids in OEt>Desat1RNAi hemolymph, particularly DAG and PE.

Thank you for the suggestion. We carried out the requested combination of DAG and PE subspecies in one figure panel each by removing the low-abundant subspecies (see new Figure 2e,f).

-Figures 4 and 5 clearly show that altered lipoprotein internalization (Fig. 4a-c) in OEt>Desat1 RNAi animals is associated with a rigid actin network and endosomal vesicles in oenocytes, and that actin suppression

treatments by overexpression of Tsr (twinstar) or Limk1 (LIM domain kinase 1) knockdown (Fig. 5a) result in a strong reduction of Apolpp accumulation on the oenocyte surface (Fig. 5b), as well as a normalization of lipoprotein levels in the hemolymph (Fig. 5c,d). What is the effect of this "rescue" of Apolpp release on starvation resistance and total TAG levels? These experiments, which are fundamental, are shown at the end of the article in Figure 7 e and f. I think the figure should be rearranged to include these results in Figure 5, as this allows a much smoother flow of results. Moreover, it would be desirable that the images' quality and the markers (Hoechst, Apolpp, and phalloidin) used in Figure 4 be reused in the immunostains shown in Figure 5.

Thank you for this suggestion. We have now moved the rescues with *LIMK1* KD and *tsr* OE from the end to Figure 5, including TAG levels and starvation sensitivity. We also increased the size of the image panels to make it reader-friendly. The resolution of the images could potentially be increased further, but for now we needed to consider the file size limits.

-This study demonstrates the relevant role of the oenocyte in the accumulation of lipids during periods of starvation, similar to that of the mammalian liver. Although this is cited in the Introduction, a paragraph highlighting the role of the oenocyte and its function as an analog of the mammalian liver, at least in fasting conditions, and interacting by signaling via ImpL2 with the fat body, should be included in the discussion.

Based on the new results, we are now discussing the role of ImpL2 in the oenocyte-fat body crosstalk in more detail. However, we believe that the paragraph regarding the analogy of oenocytes and mammalian liver is already quite long and comprehensive (page 3, line 18-27).

Reviewer #4:

In this manuscript authored by Li et al. we are presented with interesting insight into a rather complex and systemic processes that involve several organs to regulate lipid dynamic during nutrient restriction. The authors present very detailed and important description of lipidomic changes in the whole body of the fly compared to haemolymph. They demonstrate that similar as in mammals upon starvation there is a shift towards lipid desaturation and an increase in hydrocarbon length. They then investigate and confirm the participation of the oenocytes as a potential "liver-like" tissue, in addition to the fat body, in the processes that regulate the lipid shifts in haemolymph and affect fat body lipolysis. On molecular level they corroborate their findings by knocking down *Desat1*, fly desaturase, which results in the absence of starvation dependent lipid accumulation in oenocytes as well as in an increased accumulation of lipoproteins (likely derived from Fat body – not demonstrated in this paper) in intriguing rab5-positive structures wrapped in by actin at the cell cortex. The *Desat1* deficient oenocytes seem to work as a trap for lipoproteins, here demonstrated on examples of apolpp and albumin, which they demonstrate can be released by inhibiting excessive actin polymerisation. The authors looked at the lipoproteins as a likely source for lipids that may be derived from the fat body following starvation, which they assume is true based on juxtaposed timely shifts in peak lipid amounts first in FB and then in OE. Furthermore, authors proposed that the OE likely also communicate with other tissues following starvation, by releasing lipids or

proteins. They, identify Impl as one such factor and demonstrate that its release is prevented by *desat1* deficiency and rescued by inhibited actin polymerisation.

Major point. In my view this paper offers great amounts of data and delivers important evidence that oenocytes indeed contribute to lipid shifts during starvation as a liver-like tissue. While doing so the authors, provide important insights into some key molecular players and pathways that may contribute to the crosstalk with the fat body and other tissues. However, in order to claim that these molecules are indeed are either derived from the fat body or will act on the fat body, additional work must be done, i.e. using additional binary expression tools. To me it appears that the authors make many deductions when interpreting the experiments in the results section, particularly relating to OE and fat body cross talk, but not really demonstrating the validity of their statements by actually looking at the fat body or manipulating the fat body.

Thank you for this important point. Following the reviewer's suggestion, we have now employed the LexA system as an alternative binary expression system. The results are described under point #2.

The conclusions or interpretations should perhaps be revised with caution to avoid overstatements. Below, I am including some examples, excluding detailed revision of discussion:

P8 L6 The authors suggest that the apolipoproteins are FB derived. Can that be demonstrated – i.e. silencing *apolpp* in the fatbody and analysing effect on *apolpp* accumulation in OE>*Desat1R*. Alternatively, conclusion may be toned down in results section. All I see that the *apolpp* is not OE-derived.

As already illustrated in Extended Data Fig. 5b, the accumulation of *apolpp* on oenocyte surfaces following the double knockdown of *Desat1* and *apolpp* was comparable to that observed with *Desat1* knockdown alone. Furthermore, silencing of *apolpp* in oenocytes did not increase starvation sensitivity (Extended Data Fig. 5c). Collectively, these results support the notion that lipoproteins retained at the surface of *Desat1*-deficient oenocytes are not of oenocytic origin but are instead most likely fat body-derived, consistent with the literature (e.g. PMID: 22844248). The requested experiment to silence *apolpp* in the fat body in OE^{ts}>*Desat1^{RNAi}* would require an alternative binary expression system, a lengthy approach we decided to carry out only for *bmm* silencing as we feel it is more central for the storyline of the manuscript.

P10 LL4-5 “suggesting that lack of Impl2 secretion could lead to a starvation-like condition with stimulation of FB lipolysis.” – this is a valid speculation not demonstration. Therefore, statement further below in the same page LL33 does not seem valid to me: “5) Impl2 controls survival resistance by controlling FB lipolysis”. Similarly, in L35: “promoting FB lipid storage”.

Thank you for this point. In the revised version, we have included a number of new experiments that collectively strengthen the proposed role of oenocytes in the regulation of FB lipolysis. An important new experiment has been to express *bmm* RNAi specifically in the FB in OE^{ts}>*Desat1^{RNAi}* animals with the help of the LexA-LexAop system. With this approach, we could rescue starvation sensitivity and TAG levels, which clearly suggest that blocking *bmm*-dependent lipolysis in the FB is for increased survival in our prolonged starvation protocol. We now also show that *bmm* levels in both OE^{ts}>*Desat1^{RNAi}* and OE^{ts}>*Impl2^{RNAi}* flies are significantly elevated (new Fig. 3f,g and Fig. 7e), explaining why TAG levels drop so rapidly during fasting

in these animals. Furthermore, we found reduced *bmm* levels, higher TAG levels and starvation resistance in flies overexpressing ImpL2 in oenocytes (new Fig. 7e). Yet, we agree with the reviewer that the expression “starvation-like” may not be appropriate and, therefore, removed it from the paper.

P11 L 1 ie may read: “To question how exchange between BF and OE MAY BE carried out...”.

We changed this sentence accordingly.

Minor points:

1. I would like to hear more of authors thoughts on possible relationship of observed phenotypes to autophagy flux in *desat1* KD OE. To me it almost seems like the endolysosomal pathway may be stuck upon *desat1* knockdown. How specific are then those effects that we see? Does the lack of lipids perhaps perturb the membrane biogenesis and then subsequently there is a block in autophagic flux? What if the apolipoproteins are digested via autophagy after uptake into OE? And if indeed autophagy is stalled maybe that’s why we see them trapped at the surface?

Thank you for mentioning the connection between *Desat1* and autophagic flux which has been previously reported by Köhler and colleagues (PMID: 19587536). To investigate the relative contributions of autophagy versus endocytosis to the observed phenotype, we conducted the following experiments: First, we examined the subcellular localization and morphology of lysosome by expressing a GFP fused to TM domain plus C-terminus of Lamp1 construct in oenocytes (R_Fig.13). Our analysis revealed that Lamp1-GFP exhibited an altered and stronger GFP signal in the *Desat1* knockdown group compared to controls. Stronger GFP signals indicate less lysosomal degradation, suggesting that (auto)lysosome function may be compromised in *Desat1*-deficient oenocytes.

R_Fig.13 Representative immunofluorescence images of Lamp1-GFP in oenocytes with Lamp1-GFP overexpression, n=3, scale bars: 20um.

Second, to determine whether lysosomal or autophagy dysfunction alone could recapitulate the apolipoprotein particle (apolpp) accumulation phenotype, we performed targeted knockdown of *Atg1*, key regulator of lysosome and autophagosome formation, respectively, using a previously validated RNAi line (PMID: 38360932, 39455560). We hypothesized that if autophagy impairment was the primary cause of apolpp accumulation, *Atg1* knockdown should phenocopy the accumulation observed following *Desat1* knockdown. However, contrary to this prediction, apolpp localization remained unchanged in *Atg1*

knockdown oenocytes (new Extended Data Fig. 8b), suggesting that autophagy dysfunction alone is insufficient to reproduce this specific phenotype. This result is consistent with the lack of Lpp phenotype in the *Rab7* knockdown (new Extended Data Fig. 8a). Our data therefore indicate that endocytic pathway dysfunction contributes more significantly to the observed apolpp accumulation phenotype than autophagy impairment alone.

2. To me it is not clear how *brummer* contributes directly to OE phenotypes. OE>*desat1-IR*; *brummer-IR* seem to have no rescue effect, *brummer* seems to be more important in toher tissue? Additionally, *brummer* rescue experiments using the classic allele were performed on a mutant TM3 balancer background, reflected in severely reduced half life of *desat1* deficient and control flies in fig 3h compared to fig 2a,b. TM3 is a “brocken” chromosome containing multiple inversions, we don’t know how these contribute to the phenotypes studied. In order to claim a “full rescue” by haploinsufficiency in *brummer* (p7 L13), control experiments without TM3 or demonstrating *tm3* contribution to starvation sensitivity would be necessary. Same is true for the global TAG dynamics. This should be at least made more clear for educated interpretation of results by a non-Drosophilist

Thank you for raising this important question regarding the mechanism underlying lipid metabolism in *Desat1* knockdown flies.

Given our observation of significantly reduced lipid content in *Desat1* knockdown flies, we initially hypothesized that this phenotype resulted from hyperactivated *bmm* lipase activity specifically in oenocytes. To test this hypothesis, we performed double knockdown experiments simultaneously targeting both *Desat1* and *bmm* to assess whether *bmm* reduction could rescue the lipid depletion phenotype. However, our quantitative analysis of TAG levels revealed no significant difference between single *Desat1* knockdown and double *Desat1/bmm* knockdown groups. These findings indicate that enhanced lipolysis in oenocytes alone cannot account for the accelerated TAG turnover observed at the organismal level in *Desat1* knockdown flies.

To demonstrate the correlation between rapid lipid depletion and *bmm* activity, we have now performed quantitative PCR analysis on cuticle samples and detected significant upregulation of *bmm* transcript levels in both oenocyte-specific *Desat1* knockdown groups (new Figure 3f,g). To further validate *bmm*'s role for the *Desat1* KD phenotype, we performed genetic complementation using heterozygous *brummer* mutants (*bmm[1]*), exploiting haploinsufficiency to reduce *bmm* activity, as homozygous mutants are not viable. But as the reviewer rightly points out, there are important limitations in our current experimental design, such as genetic background effects and the lack of tissue-specific resolution.

To address these limitations, we embarked on a dual binary expression system approach, combining LexA-LexAop and UAS-Gal4 systems within the same animal to simultaneously manipulate *bmm* and *Desat1* expression levels in fat body and oenocytes, respectively. This approach involved cloning of new *bmm* RNAi LexAop expression vectors, generation of transgenic animals as well as testing their knockdown efficiencies, followed by lifespan experiments etc. For FB-specific expression, we used r5-LexA to drive LexAop-*bmm* RNAi in the *PromE-GAL4;tub-GAL80^{ts};UAS-Desat RNAi* background. The results nicely confirmed our previous rescue with the heterozygous *bmm[1]* mutant. Expressing *bmm* RNAi with the LexA system specifically in the fat body not only led to a suppression of the *bmm* induction in the *promE>Desat1* RNAi animals but also to significant rescue of the starvation sensitivity in these animals.

3. Regarding gene selection. P9 L24 Could the authors perhaps specify on which basis the 8 genes were selected? Additionally, extended figure 9c – it remains unclear to me why *ImpL2* was selected over the other genes – *nplp2* seems to have similar dynamic during starvation.

To comprehensively evaluate the functional roles of differentially expressed genes identified from our single-nucleus RNA sequencing analysis, we performed survival assays targeting all eight candidate genes. This systematic screening approach revealed that *dawdle* (*Daw*) and *ImpL2* exhibited the most pronounced effects on organismal lifespan. While we conducted extensive preliminary experiments on both *Daw* and *ImpL2*, we chose to focus our current manuscript on *ImpL2* due to its exceptionally clear and reproducible phenotypic profile. Specifically, oenocyte-specific *ImpL2* knockdown consistently resulted in significantly shortened lifespan during starvation conditions, while *ImpL2* overexpression produced the opposite effect—markedly extended lifespan under identical starvation stress. This robust and bidirectional relationship between *ImpL2* expression levels and survival outcomes is the main reason to focus on *ImpL2*. With regard to *Nplp2*, we did not pursue it as a primary research focus due to several considerations. First, our snRNA-seq data revealed decreased *Nplp2* expression during starvation, leading us to hypothesize that *Nplp2* knockdown would result in increased survival rates and elevated triacylglycerol (TAG) levels. However, preliminary functional validation yielded contradictory results: *Nplp2* knockdown led to reduced rather than increased TAG levels, and pilot survival experiments showed no significant differences in lifespan compared to controls. Another limitation of *Nplp2* as a candidate is its broad expression pattern extending beyond oenocytes. Unlike *ImpL2* that is specifically expressed in oenocytes during starvation, *Nplp2* is expressed across multiple tissues, making it challenging to attribute observed phenotypes specifically to oenocyte function.